# What does the mean mean? A simple test for neuroscience

**Alejandro Tlaie** [1,2]\*, **Katharine Shapcott**[1], **Thijs L. van der Plas**[3], **James Rowland**[3], **Robert Lees**[3], **Joshua Keeling**[3], **Adam Packer**[3], **Paul Tiesinga**[4], **Marieke L. Schölvinck**[1�], **Martha N. Havenith**[1,3☯]

**1** Ernst Strüngmann Institute for Neuroscience, Frankfurt am Main, Germany, **2** Laboratory for Clinical Neuroscience, Centre for Biomedical Technology, Technical University of Madrid, Madrid, Spain, **3** Department of Physiology, Anatomy, and Genetics, University of Oxford, Oxford, United Kingdom, **4** Department of Neuroinformatics, Donders Institute, Radboud University, Nijmegen, The Netherlands

☯ These authors contributed equally to this work.

\* atboria@gmail.com

## Abstract

Trial-averaged metrics, e.g. tuning curves or population response vectors, are a ubiquitous way of characterizing neuronal activity. But how relevant are such trial-averaged responses to neuronal computation itself? Here we present a simple test to estimate whether average responses reflect aspects of neuronal activity that contribute to neuronal processing. The test probes two assumptions implicitly made whenever average metrics are treated as meaningful representations of neuronal activity:

1. Reliability: Neuronal responses repeat consistently enough across trials that they convey a recognizable reflection of the average response to downstream regions.

2. Behavioural relevance: If a single-trial response is more similar to the average template, it is more likely to evoke correct behavioural responses.

We apply this test to two data sets: (1) Two-photon recordings in primary somatosensory cortices (S1 and S2) of mice trained to detect optogenetic stimulation in S1; and (2) Electrophysiological recordings from 71 brain areas in mice performing a contrast discrimination task. Under the highly controlled settings of Data set 1, both assumptions were largely fulfilled. In contrast, the less restrictive paradigm of Data set 2 met neither assumption. Simulations predict that the larger diversity of neuronal response preferences, rather than higher cross-trial reliability, drives the better performance of Data set 1. We conclude that when behaviour is less tightly restricted, average responses do not seem particularly relevant to neuronal computation, potentially because information is encoded more dynamically. Most importantly, we encourage researchers to apply this simple test of computational relevance whenever using trial-averaged neuronal metrics, in order to gauge how representative cross-trial averages are in a given context.

**Data Availability Statement:** Data are already available at https://doi.org/10.12751/g-node.h27xvl and https://figshare.com/articles/dataset/Dataset_from_Steinmetz_et_al_2019/9598406 Code is

freely available at https://github.com/atlaie/BrainAveraging/.

**Funding:** A.T. received funding (for salary) from the Margarita Salas Fellowship (NextGenerationEU) and from the Joachim Herz Stiftung. The salaries of J.M.R., R.M.L., and A.M.P. were funded by the Wellcome Trust (204651/Z/16/Z). T.L.v.d.P. acknowledges support from the Biotechnology and Biological Sciences Research Council (grant number BB/M011224/1), in the form of salary. The salaries of K.S., M.N.H. and M.L.S. were funded by the Max Planck Society. The funders had no role in study design, data collection and analysis, decision to publish, or preparation of the manuscript.

**Competing interests:** The authors have declared that no competing interests exist.

## Author summary

Neuronal activity is highly dynamic—our brain never responds to the same situation in exactly the same way. How do we extract information from such dynamic signals? The classical answer is: averaging neuronal activity across repetitions of the same stimulus to detect its consistent aspects. This logic is widespread—it is hard to find a neuroscience study that does not contain averages.

But how well do averages represent the computations that happen in the brain moment by moment? We developed a simple test that probes two assumptions implicit in averaging: Reliability: Neuronal responses repeat consistently enough across stimulus repetitions that the average remains recognizable. Behavioural relevance: Neuronal responses that are more similar to the average, are more likely to evoke correct behaviour.

We apply this test to two example data sets featuring population recordings in mice performing perceptual tasks. We show that both assumptions were largely fulfilled in the first data set, but not in the second; suggesting that the relevance of averaging varies across contexts, e.g. due to experimental control levels and neuronal diversity. Most importantly, we encourage neuroscientists to use our test to gauge whether averages reflect informative aspects of neuronal activity in their data.

## Introduction

Brain dynamics are commonly studied by recording neuronal activity over many stimulus repetitions (trials) and subsequently averaging them across time. Trial-averaging has been applied to single neurons, describing their average response preferences [1–6], and, more recently, to neural populations [7–9]. Implicit in the practice of trial averaging is the notion that deviations from the average response represent 'noise' of one form or another. The exact interpretation of such neuronal noise has been debated [10], ranging from truly random and meaningless activity [11–14], to neuronal processes that are meaningful but irrelevant for the neuronal computation at hand [15–17], to an intrinsic ingredient of efficient neuronal coding [18–20]. Nevertheless, in all of these cases a clear distinction is being made between neuronal activity that is directly related to the cognitive process under study (e.g. perceiving a specific stimulus) —which is approximated by a trial-averaged neuronal response—and 'the rest'.

While this framework has undoubtedly been useful for characterizing the general response dynamics of neuronal networks, there is a sizable explanatory gap between the general neuronal response preferences reflected in trial-averaged metrics, and the way in which neurons transmit information moment by moment. As such, using trial-averaged data as a proxy to infer principles of one-shot, moment-by-moment neuronal processing is potentially problematic—an issue that has repeatedly been discussed in the field (see for instance [21–24]). However, neuroscience as a field has so far been reluctant to draw practical consequences. A vast majority of neuroscience studies present trial-averaged metrics like receptive fields, response preferences or peri-stimulus time histograms. These metrics rely on the implicit assumption that trial-averaged neuronal activity is fundamentally meaningful to our understanding of neuronal processing. For instance, upon finding that with repeated stimulus exposure, trial-averaged population responses become more sensitive to behaviourally relevant stimuli (e.g. [3, 4]), it is implicitly assumed that this average neuronal shift will improve an animal's ability to perceive these stimuli correctly. In other words, neuroscience as a field seems to suffer from a disconnect between the limitations of cross-trial averaging that we acknowledge explicitly, and

the implicit assumptions that we allow ourselves to make when we use cross-trial averages in our work.

One potential reason that this disconnect has not been tackled more actively is that the evidence regarding the functional relevance of trial-averaged responses is quite split. On the one hand, studies highlighting the large inter-trial variability of neuronal responses [16, 17, 25–27] suggest that average responses fail to accurately capture ongoing neuronal dynamics. Then there is the simple fact that outside the lab, stimuli generally do not repeat, which renders pooled responses across stimulus repetitions a poor basis for neuronal coding. On the other hand, the fact that perceptual decisions can be altered by shifting neuronal activity away from the average response [28–31] indicates that at least in typical lab experiments [32], average population responses do matter [33]. Such widely diverging evidence suggests that cross-trial averages may be more relevant to neuronal computation in some contexts (and brain areas) than in others. This calls for a way to move the debate on their computational relevance beyond the realm of opinion and theory, and instead test this question concretely and practically across different experimental contexts.

In the present study, we provide a simple and widely applicable statistical test to explicitly determine whether cross-trial averages computed in a specific experiment are likely to be meaningful to neuronal information processing, or whether they are more likely to arise as an epiphenomenon with no clear computational function. To this end, our approach formalizes two implicit assumptions inherent in the computation of average neuronal responses, and tests directly whether they hold in a given experimental context (Fig 1). Importantly, these two testable assumptions are not based on our own or other researchers' views of how neuronal processing might actually work. Rather, they summarize how neuronal activity would need to behave if cross-trial averages reflect information that down-stream brain areas rely on to process information.

1. Reliability: The responses of task-relevant neuronal populations repeat consistently enough to recognizably reflect the average response. The rationale for this assumption is that if neuronal responses varied so widely that the trial-averaged response was in no way recognizable from single-trial responses, then that would render the trial-averaged response uninformative to downstream neuronal processing.

2. Behavioural relevance: If a single-trial response better matches the average response, it is more successful in evoking correct behavioural responses. If cross-trial averages represent the 'true signal' of a neuronal population, which is obscured by single-trial noise, then the more similar a single-trial response is to the average, the higher its signal-to-noise ratio; therefore, the easier the information readout for downstream areas; therefore, the better the chances of a successful behavioural response.

To quantify to what extent a given data set adheres to each of these assumptions, we developed two simple statistical metrics, and tested them on two complementary data sets featuring neuronal recordings in behaving mice.

## Results

We started by examining our two assumptions in a data set that was acquired under tightly controlled experimental settings. Data set 1 consists of two-photon calcium imaging recordings in primary and secondary somatosensory cortex (S1 and S2) as mice detected a low intensity optogenetic stimulus in S1 [34] (Fig 2A). Mice were trained to lick for reward in response to the optogenetic activation of 5 to 150 randomly selected S1 neurons ('stimulus present'

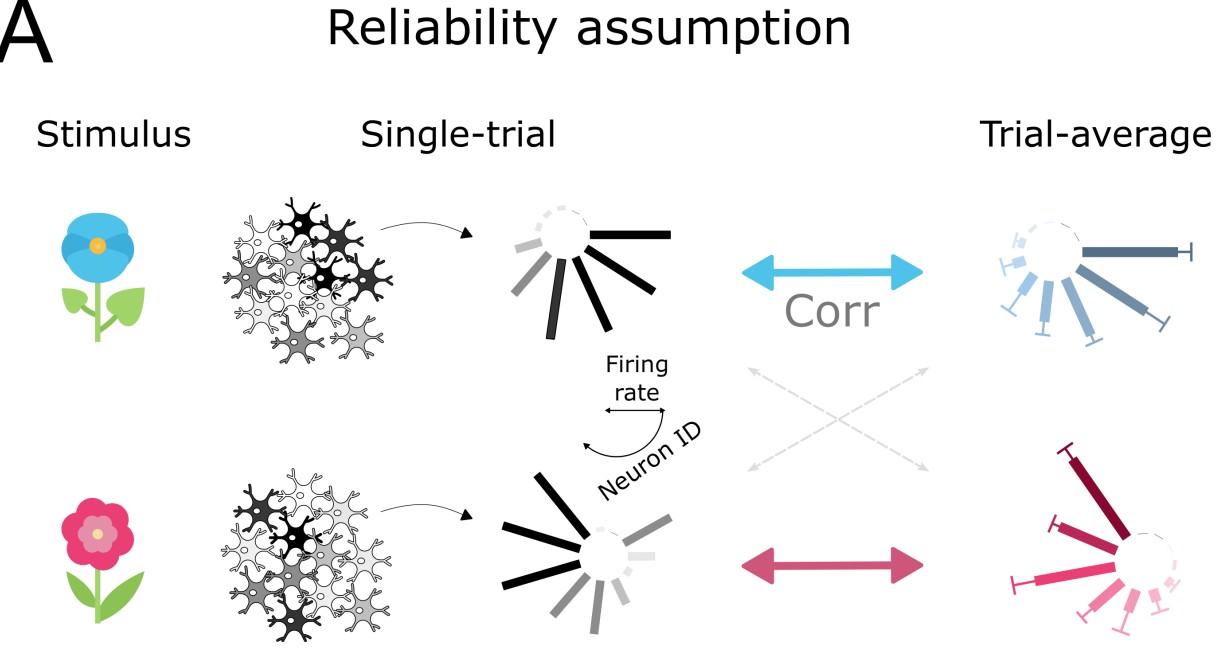

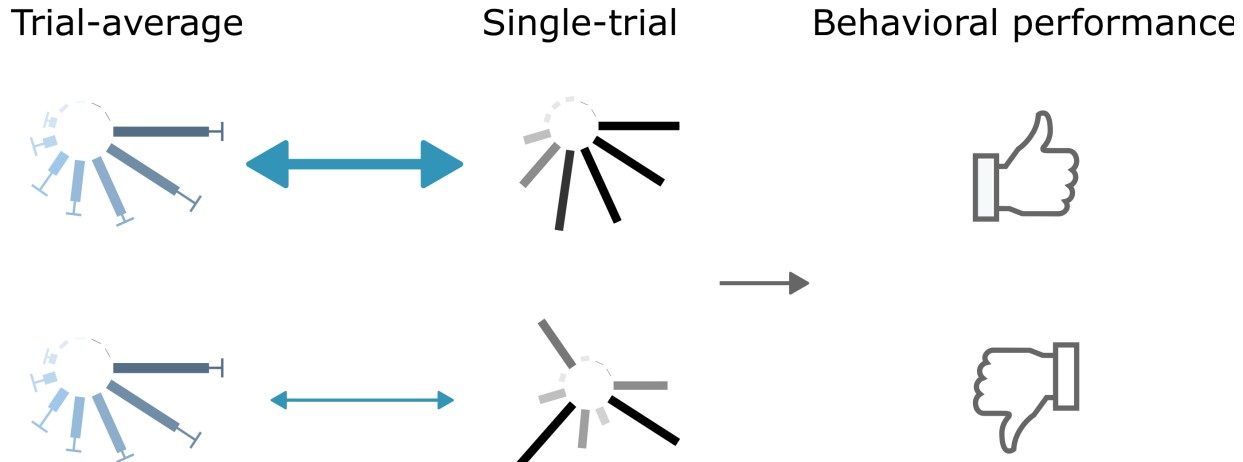

**Fig 1. Two assumptions underlying the computation of average population responses.** A) Reliability: single-trial responses correlate better with the trial-averaged response to the same stimulus, than with the trial-averaged response to a different stimulus. B) Behavioural relevance: better matched single-trial responses lead to more efficient behaviour.

condition). On 33% of trials, there was a sham stimulus during which no optogenetic stimulation was given ('stimulus absent' condition). Simultaneously, using gCAMP6s, 250–631 neurons were imaged in S1 and 45–288 in S2. Notably, in S1, the stimulus directly drives the neuronal response, skipping upstream neuronal relays.

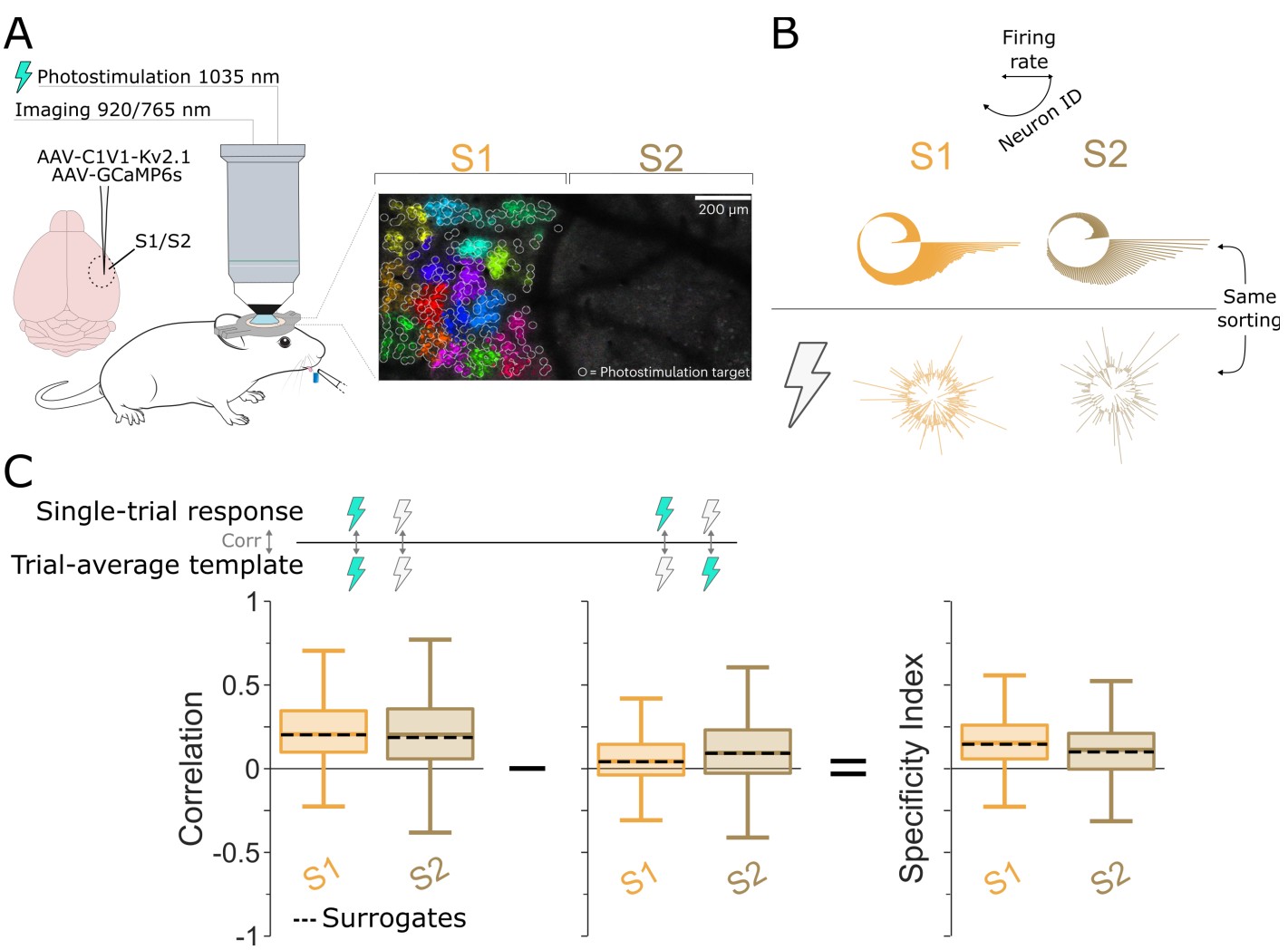

**Fig 2. Single-trial responses are stimulus-specific for Data set 1.** A) Animals report whether they perceived the optogenetic stimulation of somatosensory neurons (S1) through licking to receive reward. This panel was originally published in [34], under an open-access CC-BY license by the copyright holder. B) Trial-average population responses ('templates') for S1 (orange) and S2 (brown), under optogenetic stimulation (top) or no stimulation (bottom) conditions. Neurons are sorted the same under both conditions. C) Distribution of the correlations between single-trial responses and the matching (left) and non-matching (middle) trial-averaged response templates. Box: $25^{th}$ and $75^{th}$ percentile. Center line: median. Whiskers: $10^{th}$ and $90^{th}$ percentile. Dotted lines: median of surrogate data, which were generated by randomly sampling based on neurons' trial-averaged response probabilities for the correct template. The difference between the correlations to the matching and non-matching templates gives the Specificity Index (right).

To probe the computational role of averages within this tightly controlled setting, we first computed average population responses for the two experimental conditions. Since individual stimulation intensities were often only presented in a small number of trials, we pooled all stimulation intensities into the 'stimulus present' condition (the high correlations between the average responses to different stimulation intensities are shown in S2 Fig).

Average response *templates* were computed as the mean fluorescence ($\Delta F/F$) of each neuron in a time window of 0.5 s following the stimulation offset (Fig 2B). We next quantified how well single-trial responses matched the corresponding average template (stimulation present or absent) (Fig 2C, left; see also [35]). To this end, we computed linear correlations between the single-trial and trial-averaged population responses. While in principle, single-trial responses could reflect the corresponding average template in a multitude of ways, including

multi-dimensional and/or non-linear relations, linear correlations are the correct way to capture their match if one accepts the assumptions underlying cross-trial averaging. Averages are based on linear computations (sums and rescaling), which implicitly assumes that the single-trial responses subsumed in the average differ from each other linearly and along one dimension—otherwise, pooling them in a linear average would not be a suitable approach.

Single-trial correlations were mostly positive in both S1 and S2 (Fig 2C) ($n$ = 1795 trials; $p < 0.001$), suggesting that single-trial responses represented the average template quite faithfully. To assess if single-trial responses can be regarded as 'noisy' versions of the average template, we computed bootstrapped surrogate responses for each trial based on the neurons' average response preferences. Specifically, we created surrogate data for each trial by drawing a fluorescence value per neuron from its overall distribution of fluorescence values across all trials of one stimulus condition (for details, see S4 Fig and section Surrogate Models in Methods). The shuffled surrogate data correlated equally well to the template as the original data in both S1 and S2 (Fig 2C, left), suggesting that in Data set 1, single-trial responses can be interpreted as mostly faithful random samples of the respective average template.

Correlations between single-trial responses and the population template may partially stem from neurons' basic firing properties, which would not be task-related. To estimate the stimulus specificity of the correlations we observed, we also computed single-trial correlations to the incorrect template (e.g. 'stimulus absent' for a trial featuring optogenetic stimulation). Correlations to the incorrect template were significantly lower than to the correct one (Fig 2C, middle, Mann-Whitney U-test, $p = 5.98 \cdot 10^{-169}$, $p = 4.86 \cdot 10^{-51}$ for S1 and S2, respectively). To quantify this difference directly, we defined the *Specificity Index*, which measures, on a single-trial basis, the excess correlation to the correct template compared to the incorrect template. Thus, the Specificity Index quantifies to what extent neuronal activity in an individual trial relates to the average response of the relevant experimental condition, compared to the average responses for other experimental conditions. Since it subtracts two correlation coefficients from each other, it is bounded between -2 and 2.

Note that while single-trial correlations to the average template scale directly with the amount of inter-trial variability in a data set as well as the baseline firing rate in an individual trial, the Specificity Index is largely independent of these variables, because it reflects the differential match of single-trial responses to the correct versus incorrect template (see S6 Fig). This also means that the Specificity Index can be applied to trials that may not correlate well to cross-trial averages simply due to low spike counts. Since the correlations to two different cross-trial averages are compared to each other, higher or lower baseline correlations should not contribute to this metric.

For Data set 1, the Specificity Indices of single-trial responses indicate clear stimulus-specificity (Fig 2C, right). In addition, the observed Specificity Indices are highly similar to those reached by the corresponding surrogate data. This suggests that in Data set 1, single-trial responses can be seen as a somewhat noisy representation of the respective cross-trial average. Together, these results indicate that single-trial responses in Data set 1 were strongly and selectively correlated to the corresponding average template, largely fulfilling Assumption 1.

Next, we set out to test if the correlation between single-trial responses and average templates predicted the animal's licking behaviour (see Fig 3A, left, for an example session). To this end, we separately examined the single-trial correlations in trials that resulted in hits, misses, correct rejections (CRs) and false positives (FPs) (Fig 3A, right). For the trials where optogenetic stimulation was present, single-trial correlations in S1 were significantly higher in hit trials than in miss trials, suggesting that a better match to the average template did indeed produce hit trials more often (Fig 3B, Mann-Whitney U-test, $p = 4.96e - 68$). Similarly, while single-trial correlations were overall lower in the absence of optogenetic stimulation, correct rejections

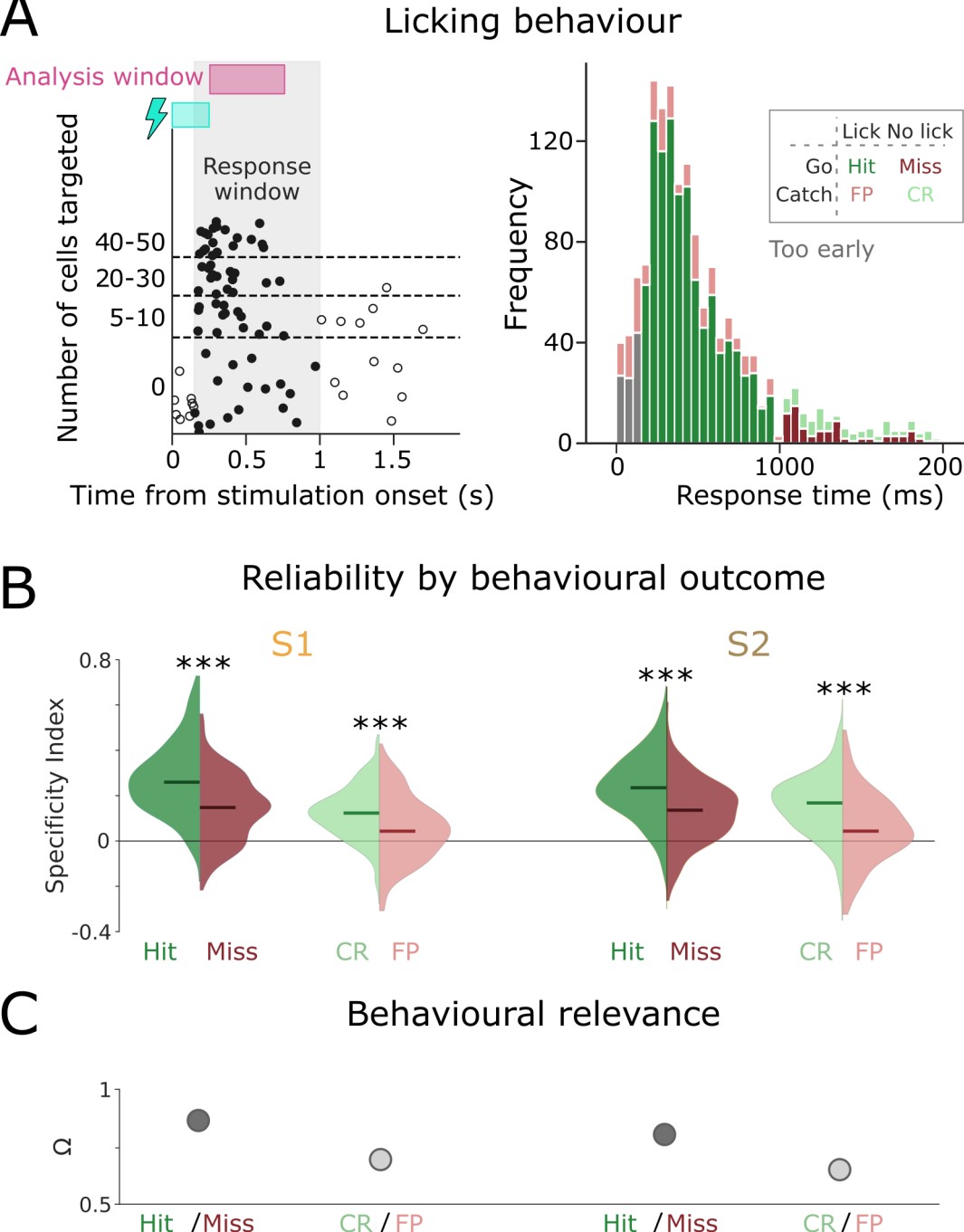

**Fig 3. Better template-matching predicts better behaviour.** A) Licking times for an example session (left) and for all sessions (right). The stimulation window is shown in blue, the analysis window in pink. This panel was originally published in [34], under an open-access CC-BY license by the copyright holder. B) Reliability of single-trial responses, as quantified by the Specificity Index, split out by hits, misses, correct rejections and false positives. C) Behavioural Relevance indices for these categories.

nevertheless featured significantly higher correlations than false positives (Fig 3B, Mann-Whitney U-test, $p = 2.82e - 17$). The same pattern held true for S2, though overall correlations were marginally smaller and the difference between correct and incorrect trials was somewhat less pronounced (Fig 3B; $p = [2.02e - 50, 1.37e - 11]$ for hit/miss and CR/FP comparisons, respectively). To quantify directly to what extent single-trial correlations predicted behaviour, we computed the *Behavioural Relevance Index* ($\Omega$) as $\Omega = max(A, 1 - A)$, where $A$ is the Vargha-Delaney's effect size [36] (see the section on Specificity and Behavioural Relevance Index in Methods). The Behavioural Relevance Index quantifies whether successful behavioural responses occur preferentially in trials with a higher Specificity Index. The Behavioural Relevance Index is bounded between 0.5 and 1, with 0.5 indicating complete overlap between the distributions of Specificity Indices for correct and incorrect trials, and 1 meaning no overlap at all. For both the trials with stimulation (hits and misses) and without stimulation (CRs and FPs) $\Omega$ exceeded 0.5 in S1 and S2 (Fig 3C). This suggests that in both areas, single-trial responses that were better matched to the corresponding cross-trial average resulted in more successful behaviour, fulfilling Assumption 2. Together, these results indicate that in Data set 1, cross-trial averages are both reliable and behaviourally relevant enough be computationally meaningful.

Building on these results, we set out to determine how computationally meaningful cross-trial averages might be within the less restrictive experimental paradigm of Data set 2. Data set 2 contains high-density electrophysiological (Neuropixel) recordings across 71 brain regions (Fig 4A, right) in mice performing a two-choice contrast discrimination task [31]. Mice were presented with two gratings of varying contrast (0, 25, 50 or 100%) appearing in their left and right hemifield. To receive reward, animals turned a small steering wheel to bring the higher-contrast grating into the center, or refrained from moving the wheel if no grating appeared on either side (Fig 4A, left). When both stimulus contrasts were equal, animals were randomly rewarded for turning right or left. Those trials were discarded in our analysis since there is no 'correct' behavioural response.

Since this data set contains neuronal recordings from 71 brain areas, not all of which may be directly involved in the perceptual task at hand, we used a data-driven approach to identify to what extent neuronal population activity predicted the presented stimulus and/or the animal's target choice. We trained a decoder (Multinomial GLM, see section Decoders in Methods) based on single-trial population vectors, to identify either target choice (left/right/no turn) or stimulus condition (higher contrast on left/right, zero contrast on both). For the neuronal response vectors, we considered neuronal activity $0 - 200ms$ post-stimulus onset (S5 Fig). We then computed the mutual information between the decoder predictions and the real outcomes (Fig 4B; see section Decoders in Methods).

Many brain areas appeared to contain little task-relevant information (shown in black in Fig 4B). We therefore used a standard elbow criterion (see section Decoders in Methods) to determine a threshold for selecting brain areas that provided the highest information on either stimulus ($I_{stim}^{thr} = 0.242$ bits; blue areas), choice ($I_{choice}^{thr} = 0.248$ bits; red areas), or both (i.e. both thresholds exceeded; purple areas). These areas seem largely congruent with the literature. For instance, primary visual cortex (VISp) is expected to reflect the visual stimulus, while choice information is conveyed e.g. by the ventral anterior-lateral complex of the thalamus (VAL)—known to be a central integrative center for motor control [37]. As an example of a both choice- and stimulus-informative area, we see caudoputamen (CP)—involved in goal-directed behaviours [38], spatial learning [39], and orienting saccadic eye movements based on reward expectancy [40].

In principle, the selection of relevant brain areas might be dependent e.g. on the specific cut-off thresholds introduced by the elbow criterion we applied. To test the validity of this

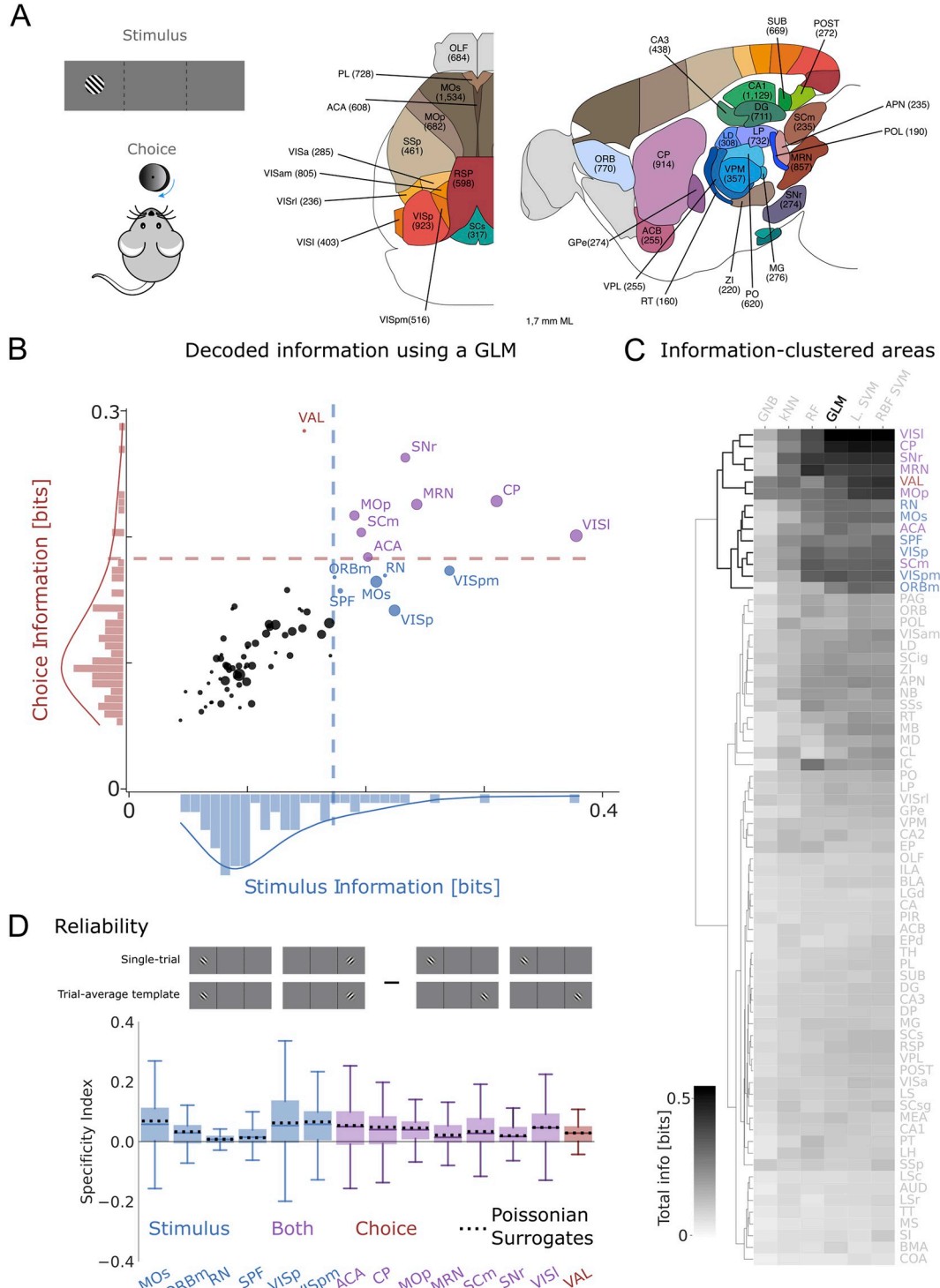

**Fig 4. Single-trial responses are hardly stimulus-specific for Data set 2.** A) Graphic representation of paradigm used in Data set 2. Animals move a steering wheel to move the higher-contrast grating of two alternative grating stimuli towards the centre (left), while being recorded from 71 brain areas (right). Note that grating depicted here does not accurately represent the grating stimuli used. B) Stimulus and target choice information decoded by a multinomial GLM decoder (Methods) from the neuronal activity in all recorded brain areas. Each point represents the median (dot location) and standard deviation across sessions (dot size) of one brain area (see in-figure labels). Colours (blue, red, purple) represent those areas where (stimulus, choice, both) information was above an elbow criterion. C) We repeated the decoding with other models (see labels) and then performed a hierarchical clustering of the total mutual information of the ranked brain areas (rows). The 14

areas we found with the GLM (see B) are consistently found with other decoders. D) Specificity Index of the selected areas, defined as the difference in the correlations between single-trial responses and the matching (cartoon, left) and non-matching (cartoon, right) trial-averaged response templates. Box: 25$^{th}$ and 75$^{th}$ percentile. Center line: median. Whiskers: 10$^{th}$ and 90$^{th}$ percentile. Shaded areas: 5$^{th}$ and 95$^{th}$ percentiles of bootstrapped data. Dotted lines: median Specificity Index for the bootstrapped surrogate data, which were generated for each recorded area using Poissonian sampling of the trial-averaged response templates.

selection process, we repeated the analysis with five other decoders (Fig 4C). We then ranked the total amount of Mutual Information per area (stimulus + choice information) for each of these models. Finally, we performed a hierarchical clustering to determine areas that were consistently classed as highly informative across decoders. Interestingly, the brain areas identified via the elbow criterion in Fig 4B coincided exactly with the top performing cluster in our multi-decoder analysis (Fig 4C). As such, both analyses converged on a group of 14 brain areas that conveyed significant task information regardless of decoder approach.

Having identified task-relevant brain areas, we used neuronal recordings from those informative areas to test the two assumptions set out in the Introduction. First, we computed the average population response templates for different experimental conditions. To avoid working with trial numbers as low as $n = 2$ for specific contrast combinations, we pooled several contrast levels requiring the same behavioural response (e.g. 50% right—0% left and 100% right—50% left) into two conditions: Target stimulus on the left or on the right. Average responses to the individual contrast levels were very comparable (S2 Fig).

To test the first assumption, as we did for Data set 1, we quantified how well single-trial responses correlated with the average template for a given stimulus (S3 Fig (A); see also [35]). Median correlations ranged from $r = 0.56$ to $0.89$ across all brain areas ($n = 89$ to $3560$ trials per brain area; all $p < 0.001$), suggesting that single-trial responses clearly resembled the average template. As a control, we computed 100 bootstrapped response vectors for each trial. Specifically, for each trial we generated the same number of spikes as recorded in the original trial, and randomly assigned these spikes to the various neurons in the population based on the probabilities given by their average firing rates (S3(B) Fig; see section Surrogate Models in Methods). Surrogate data created in this way should converge towards the original data if trial-averaged firing rates represent the true response, which is sampled discretely via a Poisson process in individual trials. These Poissonian surrogate data uniformly correlated better to the template than the original data (S3(A) Fig). In other words, single-trial responses in Data set 2 exhibited more variation than explained simply by (Poissonian) down-sampling of the firing preferences represented by the average template into a specific number of discrete spikes in an individual trial. As in Data set 1, single-trial correlations scaled with the amount of inter-trial variability, reflected for instance by the Fano Factor (S6 Fig). Note that the Fano Factors for all analysed brain areas in Data set 2 fell within the range of previously reported results [41, 42] (S6(B) Fig), suggesting that Data set 2 provides a representative example for neuronal activity in these brain areas.

Next, we estimated the stimulus specificity of the observed correlations by computing single-trial correlations to the incorrect template (e.g. 'target right' for the left target; see Fig 4D, top). These were broadly distributed, but on average marginally lower than the single-trial correlations to the correct template (S3(B) Fig). Consequently, Specificity Indices across all 14 brain areas were mostly positive but rarely exceeded 0.1 (Fig 4D, bottom; note that the Specificity Index is bounded by −2 and 2). In other words, correlations between single-trial responses and template were largely stimulus-independent. This lack of response specificity was not directly predicted by the amount of inter-trial variability, because the Specificity Index

reflects the differential link to correct versus incorrect template rather than the overall reproducibility of neuronal responses (see S5(B) Fig).

These results tally with recent work demonstrating how strongly non-task-related factors drive neuronal responses even in primary sensory areas like visual cortex [16, 17, 43–47]. However, despite these factors, the animal still needs to arrive at a coherent perceptual choice (e.g. steering right or left, see Fig 5A)—and indeed succeeds in doing so in most trials. To test if trial-averaged templates are relevant to this perceptual decision, we compared single-trial correlations for hit trials (correct target choice) and miss trials (incorrect target or no response). Single-trial correlations were marginally lower in miss trials than in hit trials across most brain areas (Fig 5B). However, their difference was small, leading to Behavioural Relevance Indices between 0.51 and 0.66 (where the Behavioural Relevance Index is bounded between 0.5 and 1). According to the Vargha & Delaney's effect size, such values would be considered largely negligible, indicating that single-trial correlations are not a reliable way to predict subsequent behaviour in Data set 2 (Fig 5C; see also [36]).

Together, these results suggest that in Data set 2, the relation between single-trial responses and trial-averaged firing rate templates was only marginally stimulus-specific, and did not appear to substantially inform subsequent behavioural choices. However, this estimate may present a lower bound for several reasons. First, while the task information conveyed by cross-trial averages seemed to be limited in the recorded population of neurons, it might be sufficient to generate accurate behaviour when scaled up to a larger population. To explore this possibility, we sub-sampled the population of recorded neurons in each brain area from $N/10$ to $N$. We then extrapolated how Specificity and Behavioural Relevance would evolve with a growing number of neurons. These extrapolations indicated that taking into account larger neuronal populations seemed to be at least somewhat beneficial for the Specificity and Behavioural Relevance of cross-trial averages in Data set 2, though improvements were rather moderate (Fig 6A, right). This did not seem to be an inherent feature of our extrapolation approach: for Data set 1, the Specificity Index appeared to remain largely stable with growing $n$, but $\Omega$ rose steeply, indicating that with more neurons, single-trial correlations to the average template would more robustly predict behaviour (Fig 6A, left).

Alternatively, at least in the brain areas most involved in task processing, there might be a group of 'super-coder' neurons that reflect relevant task variables more consistently [48, 49]. To test this possibility, we implemented a jackknife procedure, removing one neuron at a time from the data and recomputing all metrics based on the remaining population. This approach generally did not reveal neurons that particularly boosted single-trial correlations or Specificity (S9 Fig). Rather, the contribution of different neurons to the population response's Specificity was distributed largely symmetrically around zero (as measured by $\gamma$; S9 Fig lower panels). However, a few brain regions in Data set 2 did feature a somewhat right-skewed distribution of jackknifed Specificity Indices (S9 Fig, indicating that for these areas, at least some neurons contributed substantially to single-trial correlations. Intriguingly, such super-coder neurons contributed more heavily to the Specificity Index in hit than in miss trials. This suggests that when super-coder responses were more specific, animals tended to act more successfully. The areas highlighted by this analysis are consistent with the notion that 'super-coder' neurons might appear in brain areas most directly involved in task processing. Specifically, the analysis identified six brain areas: The red nucleus (RN, a subcortical hub for motor coordination), the subparafascicular thalamic nucleus (SPF, auditory processing), the primary visual cortex (VISp, visual processing), reticular substantia nigra (SNr, reward and motor planning) and midbrain reticular nucleus (MRN, arousal and conscious state).

In contrast, in Data set 1, we found no evidence of super-coder neurons altogether, i.e. $\gamma \approx 0$. This may reflect the fact that task information is distributed evenly throughout the neuronal

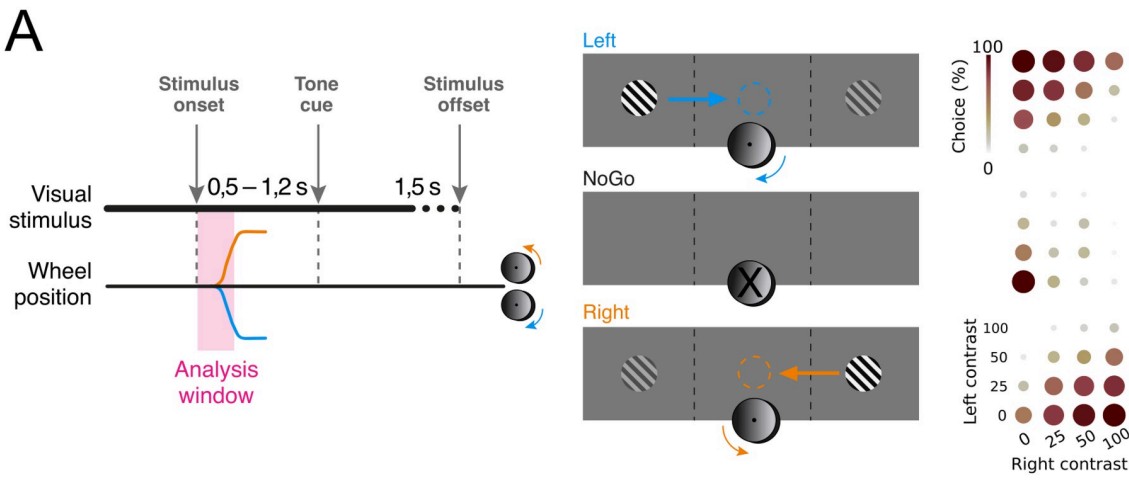

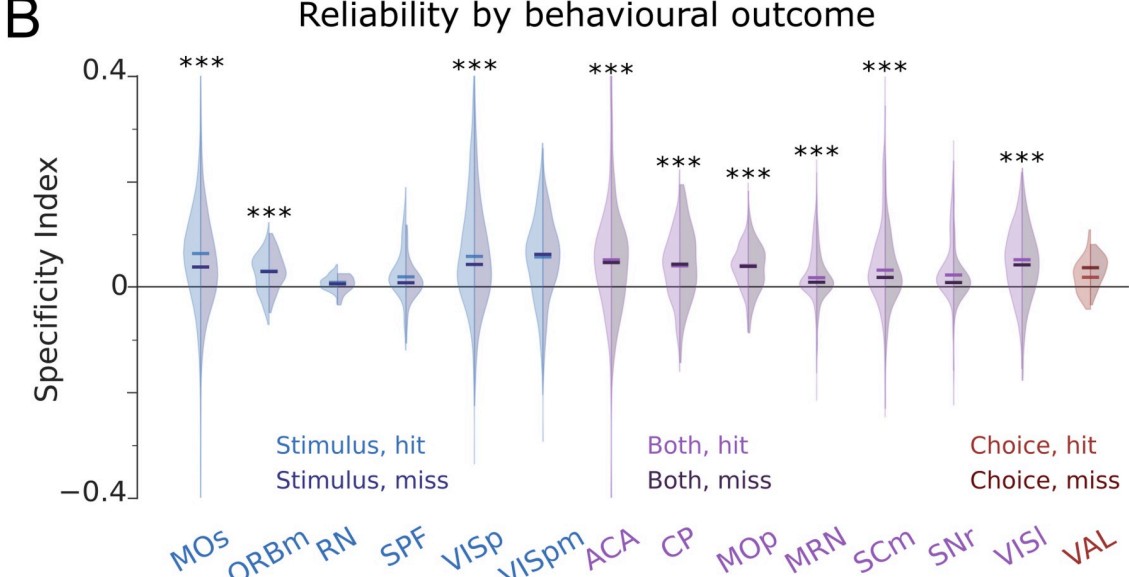

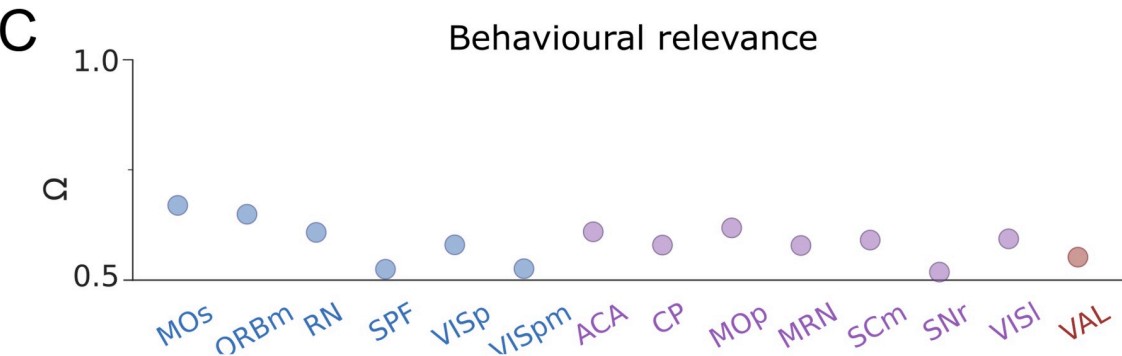

**Fig 5. Single-trial responses are barely behaviourally relevant for Data set 2.** A) Graphic representation of task structure in Data set 2. Note that gratings depicted here are not accurate representations of the grating stimuli used. Mice move the higher contrast grating stimulus towards the centre by steering a wheel, or refrain from moving the wheel when no stimulus is present (left, middle). Animals accomplish this task with high proficiency (right). We show representations of the stimuli instead of the actual gratings. B) Specificity Index for the selected areas, split by hits and misses. C) Behavioural Relevance for selected brain areas.

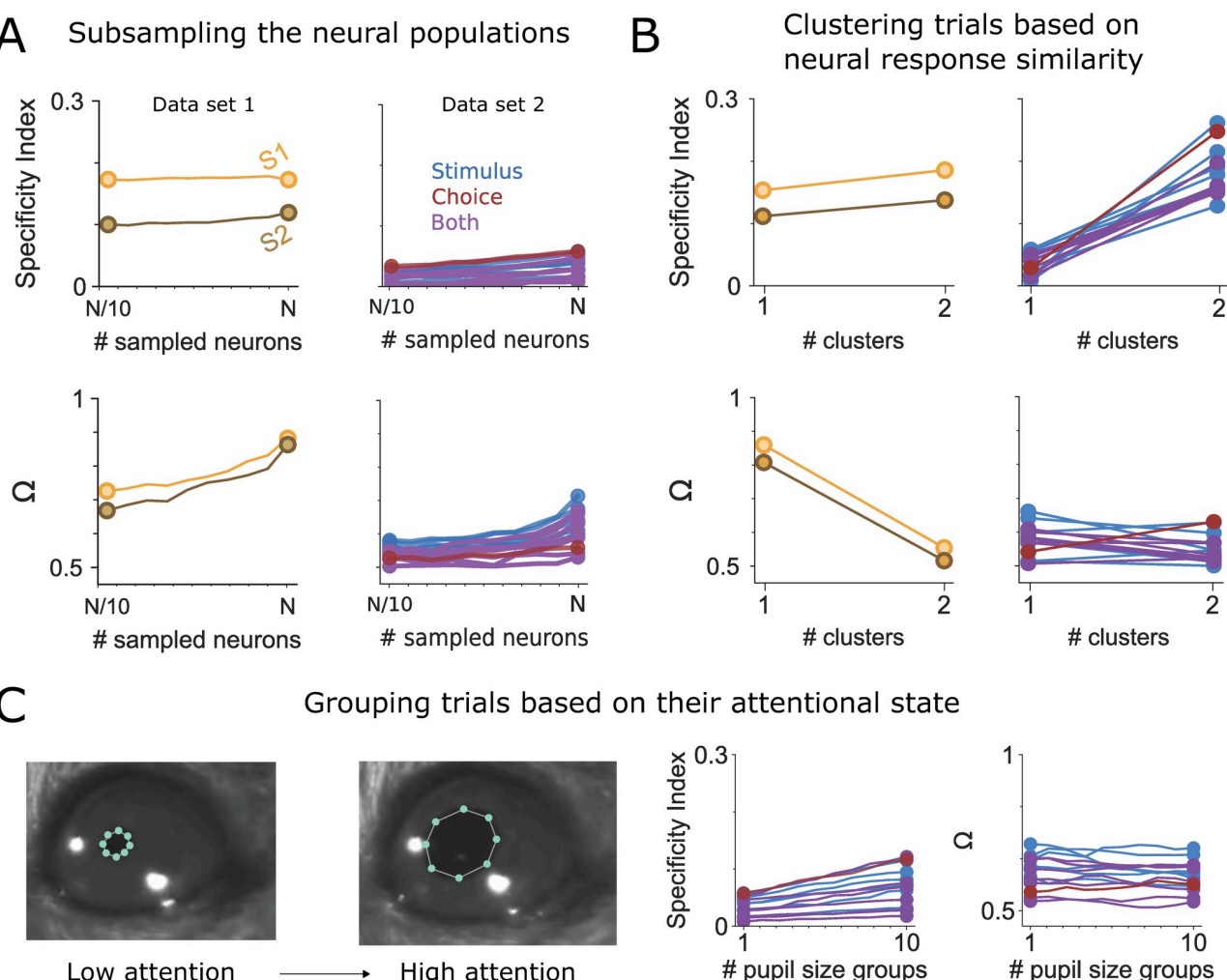

**Fig 6. Control analyses for both data sets.** A) We subsampled the neuronal populations to check whether we could extrapolate a marked benefit from adding neurons when performing template-matching. Increasing the number of sampled neurons left the Specificity Indices for both Data sets largely unchanged (top), and yielded slight increases in Ω (bottom). We then clustered trials based on the similarity in their neural response (B) and pupil size (C). These clusterings had either a negative (Data set 1) or no (Data set 2) effect on the Behavioural Relevance Index, and only slightly increased the Specificity Index for both data sets.

population, or that Data set 1 did not contain recordings from brain areas that might contain 'super-coder neurons'. Together, these results imply that while there were some neurons whose averaged responses reflected task-relevant information more robustly, these neurons were rather rare, and far from perfectly reliable. In other words, in both data sets, any sub-set of neurons could likely convey the average response template to approximately the same extent under most circumstances.

Even if population responses generally did not feature a clearly distinct group of neurons with particularly reliable responses, it is in principle possible that downstream areas only 'listen to' the neurons at the most informative tail of the distribution, and ignore responses from less informative neurons. To explore whether in this scenario, single-trial responses would clearly reflect the relevant cross-trial averages, we sub-sampled of each neuronal population to include only the 10 percent of neurons that had emerged as most and least stimulus-specific,

respectively, based on the jackknifing procedure detailed above. As one would expect, the Specificity Index derived from the most reliably stimulus-selective neurons was generally higher than that of the least informative neurons, even though the difference was reasonably small ($\Delta$Specificity Index $< 0.2$; see S10 Fig). In contrast, Behavioural Relevance ($\Omega$) decreased in most regions when only a sub-group of neurons was considered—whether most or least informative (S10 Fig). This suggests that cross-trial averages based only on the individually most informative neurons failed to reflect behaviourally relevant information more successfully than those based on a wider range of neuronal responses.

In addition, by pooling stimulus pairs with large and small contrast differences into just two stimulus categories—'target left' and 'target right'—we may have caused the resulting average templates to appear less distinctive. Specifically, difficult stimulus pairs might 'blur the boundaries' between average templates. To estimate the impact of stimulus similarity on the Specificity Index and Behavioural Relevance, we computed them separately for difficult and easy trials. Both Specificity and Relevance increased in a majority of brain areas when only taking into account stimulus pairs with large contrast differences, but plummeted for more subtle contrast differences (see S8 Fig). This suggests that in Data set 2, single trial population firing rates were both specific and behaviourally relevant when processing coarse stimulus information, but only barely in finer contrast discrimination. In this context, it is important to note that animals were also highly successful in discriminating difficult stimulus pairs. This suggests that fine contrast discrimination relied on other coding modalities than average firing rates.

Another potential limiting factor of our analysis could be that template-matching may occur in a way that cannot be captured by simple correlations. Even though linear correlations are in principle the correct way to test the assumptions inherent in cross-trial averaging, it is still possible that the linear operation of cross-trial averaging somehow reflects neuronal features of single-trial responses that are better understood in a higher-dimensional space. To explore this scenario, we repeated all previous analyses, but characterized population responses using Principal Component Analysis (PCA) via Singular Value Decomposition (SVD), and quantified their resemblance (normalized distance, see section PCA Analysis in Methods) to the average template in this lower-dimensional space. In both data sets, stimulus specificity increased marginally, but behavioural relevance decreased (S12 Fig and Fig 7). The decrease in behavioural relevance was particularly steep in Data set 1, indicating that raw firing rates were more instructive to the animals' choices than lower-dimensional features of neuronal activity (Fig 7; for details, see S12 Fig).

Finally, neuronal responses may reflect a multi-factorial conjunction of response preferences to a wide range of stimulus and behavioural variables. To test this hypothesis, we quantified whether single-trial correlations to the average would become more specific or behaviourally relevant when additional variables were taken into account. As a first test, we accounted for potential modulating variables in an agnostic way by clustering the neuronal population responses from all trials according to similarity. Such clusters might reflect different spontaneously occurring processing states that the animal enters into for reasons (e.g. locomotion, satiation, learning etc.) that may remain unknown to the experimenter. Based on the Silhouette Index, which measures cluster compactness (S11 Fig), we decided to group trials into two clusters. We repeated all analyses of Specificity and Behavioural Relevance within each of these trial clusters. In Data set 1, the Specificity Index was largely unchanged (Fig 6B; see Fig 7 for a summary). This aligns with the fact that clusters in Data set 1 were less compact and thus trial-grouping did not significantly reduce response spread (S11(A) Fig). At the same time, Behavioural Relevance decreased rather sharply (Fig 6B), indicating that differences between the identified trial clusters were in fact behaviourally informative.

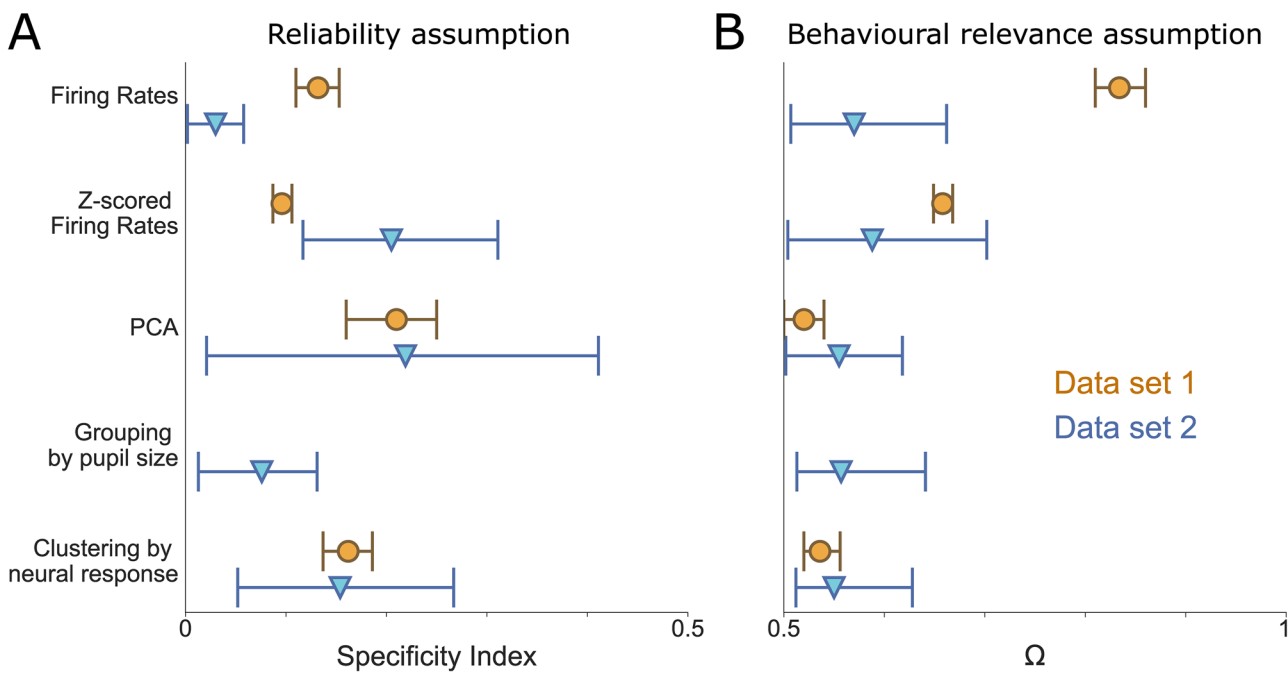

**Fig 7. Summary of the results in both Data sets.** The Y-axis shows the various methods used before Specificity (left) and Relevance (right) Indices are computed.

In contrast, Specificity rose sharply in Data set 2 after trial clustering (Fig 6B; see Fig 7 for a summary). This suggests a scenario in which the same stimulus can evoke multiple distinct, but self-consistent, response patterns. These patterns are obscured when responses are averaged across all trials indiscriminately, but emerge robustly when averages are computed separately per trial cluster. However, the lack of improvement in Behavioural Relevance suggests that single-trial correlations to the average did not consistently predict behaviour in either response mode. The test presented here can help researchers to reveal and explicitly address such unique neuronal response dynamics.

As a second test, we accounted for spontaneous fluctuations of attentional state as reflected by pupil size [45, 50, 51]. To this end, we grouped trials by pupil size and computed cross-trial averages for these trial groups. If population responses are modulated by attentional state, examining only trials that occurred during similar attentional states should reduce unexplained variability. However, grouping trials by average pupil size only slightly improved Specificity, and did not improve Behavioural Relevance at all in Data set 2 (Fig 6C; Data set 1 did not contain measurements of pupil size).

Together, these analyses suggest that the missing link between single-trial responses and cross-trial averages in Data set 2 is not sufficiently explained by unmeasured confounding factors, non-linear interactions or lack of neurons. Rather, it appears to be an inherent feature of the data set. Fig 7 summarizes the outcomes of different analysis approaches. Across analyses, Data set 1 generally shows better Specificity and Behavioural Relevance than Data set 2. Interestingly, the Specificity in Data set 1 did not increase further with procedures such as Z-scoring neuronal responses over trials to remove baseline firing rates, applying PCAs to extract lower-dimensional response features or clustering trials according to similarity—and Behavioural Relevance was even reduced by these procedures. This suggests that in Data set 1, absolute

firing rates were instructive of behaviour on a single-trial level, so that metrics obscuring absolute firing rates (e.g. by dimensionality reduction) impaired their Behavioural Relevance. This finding is particularly interesting given that in this paradigm animals had essentially been trained to detect extra spikes in somatosensory cortex. As such, it seems plausible that the neuronal representation of this task would feature absolute spike counts.

In contrast, in Data set 2, response specificity clearly benefited when baseline firing rates were accounted for—either by z-scoring firing rates, applying PCA, or clustering trials according to similarity. This suggests that here, absolute firing rate fluctuations were not tied to task performance, so that removing such fluctuations helped to uncover task information available at the single-trial level. However, this did not improve Behavioural Relevance. Thus, while baseline firing rates might have obscured the reliability of single-trial responses, removing them did not produce more behaviourally relevant cross-trial averages.

These results raise several questions. What features of Data set 1 make cross-trial averages so much more representative of single-trial processing than in Data set 2? And how representative are those features compared to the range of data generated by neuroscience? To start delineating answers to these questions, we created a simple model to simulate what the Specificity and Behavioural Relevance Index would be when our tests were applied to neuronal population responses with (1) different distributions of response preferences, (2) different degrees of neuronal single-trial variability and (3) different degrees of variability in the translation from neuronal responses to behavioural choice. Specifically, we simulated a neuronal population of similar size as those recorded in both data sets ($n = 200$). For these 200 neurons, we simulated a $Beta(\beta, \beta)$ distribution, governed by a parameter $\beta$, of average response preferences regarding two hypothetical stimuli. When $\beta = 1$, response preferences were distributed completely uniformly across the spectrum from Stimulus 1 to Stimulus 2. $\beta < 1$ indicated a shift towards an increasingly segregated bimodal distribution, with neurons preferring either Stimulus 1 or 2. $\beta > 1$ indicated an increasingly tight unimodal Gaussian function (even though the Beta distribution does not exactly equate the Gaussian distribution for any set of parameters, in the case of $\lim_{\alpha \to \infty} \beta(\alpha, \alpha)$, the Beta function converges to the standard normal distribution [52]), with all neurons responding in largely the same way regardless of the presence of Stimulus 1 or 2. Next, single-trial responses of each simulated neuron were the sum of its 'true' response preference, and a varying level of 'noise'. Based on these simulated single-trial responses, we modelled a behavioural read-out that acted as a non-linearity, choosing the correct or incorrect behavioural output depending on how similar a given neuronal single-trial response was to the correct versus incorrect average template (see section on Simulations in Methods). This non-linearity could either act in a noise-free manner, translating single-trial responses directly to the most closely corresponding behavioural decision, or it could add some 'decision variability' of its own (Fig 8A).

These simulations showed that the Specificity Index will rise steeply when the distribution of neuronal response preferences is reasonably spread out ($\beta <= 1$). In contrast, the Specificity Index will remain low as soon as neuronal response preferences are not particularly stimulus-specific ($\beta > 1$). Interestingly, this overall pattern was only marginally dependent on the amount of single-trial 'noise' added to the simulated average response preferences (Fig 8B). In other words, the diversity of neuronal response preferences was much more crucial to the computational utility of cross-trial averages than low cross-trial variability. Comparing the median Specificity Index of Data sets 1 and 2 to these simulations (Fig 8B) suggests that in order to reach the Specificity values we observed empirically, neuronal responses in Data set 1 should be distributed at least somewhat bimodally between the two stimulus conditions, while those in Data set 2 should be less distinguishable from each other. The real distributions of neuronal response preferences in both data sets confirmed these predictions (S14 Fig). These

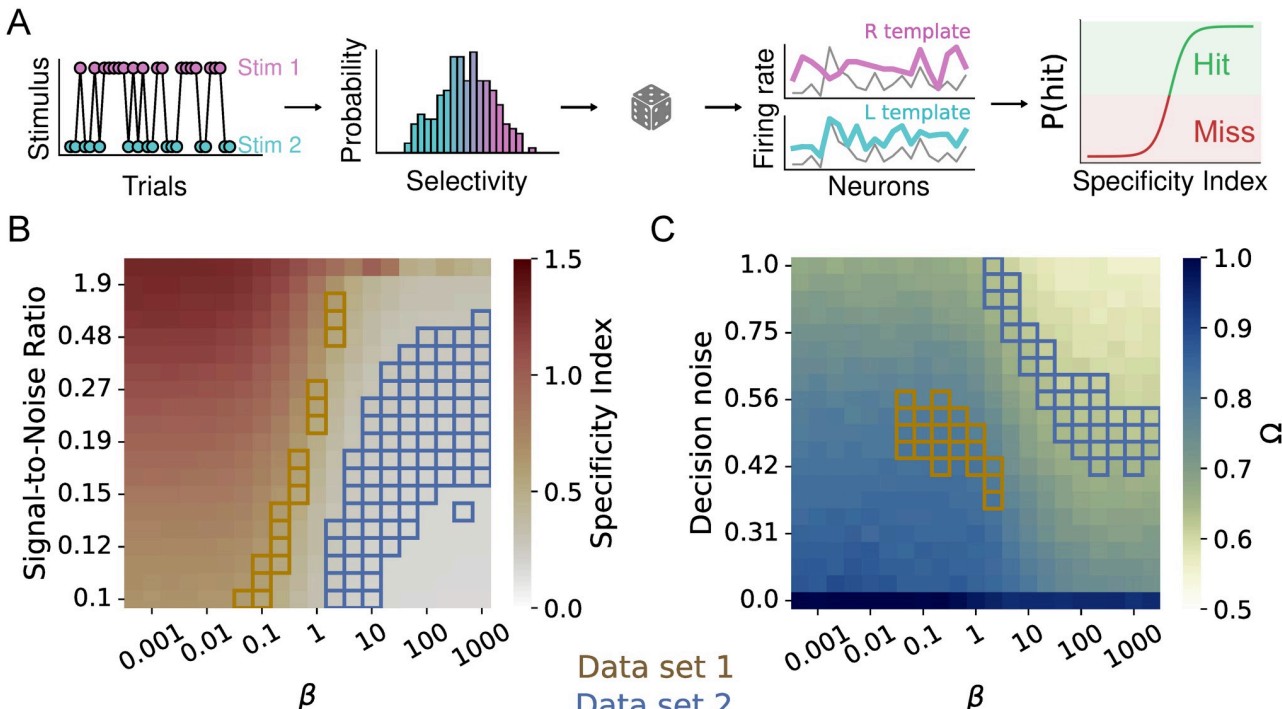

**Fig 8. Template-matching simulation.** A) Stimuli are randomly sampled from a bimodal distribution. Then, neural responses are modelled as a baseline firing rate plus a stimulus-related response (with a background noise term, parametrized by an SNR), modulated by the selectivity $\beta$. Finally, the choice is made by passing the Specificity Index (difference between the correlation to one stimulus minus the other) through a sigmoid with Gaussian noise. (B) Specificity Index as we vary the SNR and the selectivity ($\beta$) of the model neurons. We have highlighted the points in the simulation that are compatible with the experimental data sets (color coded as indicated in the legend). Compatibility is defined by a threshold of $|SpecIdx_{measured} - SpecIdx_{model}| < 0.05$. (C) Same as B), but for the Behavioural Relevance Index ($\Omega$). In this case, we varied the noise intensity of the decision-making process, for a fixed intermediate SNR.

simulations therefore suggest that the improved Specificity of cross-trial averages in Data set 1 compared to Data set 2 largely hinges on the broader distribution of neuronal response preferences.

We next explored how the Behavioural Relevance Index reflected the interplay between the distribution of neuronal response preferences, and the variability in the decision making process itself. Like the Specificity Index, Behavioural Relevance increased with a broader distribution of response preferences, as well as (unsurprisingly), with smaller variability of decision making (Fig 8C). Based on their median Behavioural Relevance Indices, our two example data sets would appear to occupy distinct areas of the parameter landscape: Based on its measured Behavioural Relevance, Data set 1 seems to operate in a regime with an at least somewhat bimodal distribution of neuronal preferences and a moderate variability of behavioural decision making, meaning that single-trial correlations to the average template drive behavioural decisions quite faithfully. In contrast, Data set 2 would be predicted to feature either a tight unimodal distribution of neuronal response preferences, combined with similarly faithful decision making as Data set 1—or a more spread-out, uniform distribution of neuronal preferences, but with extremely noisy decision making. Given the response distributions shown in (S14 Fig), we assume that both unspecific neuronal response preferences and genuine decision-making variability contribute to the low Behavioural Relevance observed in Data set 2.

Together, these simulations demonstrate three main outcomes. First, the Specificity and Behavioural Relevance Index are able to correctly pick up the neuro-behavioural features they

were designed to reflect—stimulus-specific neuronal response profiles in the case of the Specificity Index, and consistent decision criteria based on these neuronal response profiles in the case of the Behavioural Relevance Index. Second, our two example data sets occupy distinct and complementary spaces within the parameter landscape of neuronal and behavioural variability, with the lower effectiveness of cross-trial averages in Data set 2 being most likely due to a tighter distribution of neuronal preferences and higher variability of decision making. Finally, based on the Specificity and Behavioural Relevance computed in real data, simulations like the one presented here can in fact be used to hypothesize about the neuronal and behavioural mechanisms that boost or bust the computations relevance of cross-trial averages in a specific experiment. For instance, our simulation highlighted the broad distribution of neuronal response preferences, rather than the magnitude of single-trial variability, as the main factor that makes cross-trial averages more meaningful in Data set 1 than in Data set 2.

## Discussion

The present study set out to formulate an explicit statistical test to determine how reflective average population responses are of neuronal processing in different contexts. To this end, it posits two testable assumptions that should hold if cross-trial averages are computationally relevant: 1) single-trial responses should be sufficiently reliable to resemble the correct average template, and 2) better-matching single-trial responses should evoke more efficient behaviour.

To directly quantify to what extent these two conditions are fulfilled for a given data set and averaging approach, we introduced two simple metrics: (1) the Specificity Index, which captures whether a single-trial response is more related to the average response of the true experimental condition than to the average responses of other experimental conditions; and (2) the Behavioural Relevance Index, which reflects whether higher single-trial correlations to the correct average template result in more behaviourally successful trials.

With this test, we aim to bridge an in our view crucial gap in the way neuroscience is currently practiced: The disconnect between our explicit knowledge that neuronal population activity typically evolves continuously according to non-linear dynamics, which are often poorly captured by cross-trial averaging ([25, 53–56]), and the implicit assumptions we accept when nevertheless treating cross-trial averages of neuronal activity as an informative summary metric. Our two metrics are easy to compute and thereby allow researchers to explicitly determine whether their trial-average metrics are likely to be sufficiently reliable and behaviourally relevant to play a functional role in a given experimental context.

To establish the general applicability of our metrics, we tested them on two complementary data sets, featuring tight experimental control of neuronal stimulation and highly trained, simple behavioural responses in the first data set, and less tightly controlled visual stimulation with a more variable and naturalistic behavioural output in the second data set. The two assumptions of cross-trial averaging were largely fulfilled in only the first data set, which is surprising because in this experiment, optogenetic stimulation targeted a somewhat overlapping but randomly selected population of 5 to 150 neurons in each new trial. Thus, even though optogenetic stimulation varied randomly, it seemingly managed to recruit a reproducible network of neurons within the analysis time window of 500 ms post-stimulation. This suggests that the population responses highlighted by our analyses of Data set 1 rely on 'hub neurons' that are activated by various different stimulation patterns. Moreover, this is in line with the encoding of stimulus responses across trials by a consistently weighted set of neurons [34].

In contrast, in Data set 2, single-trial correlations were lower than expected from a Poissonian spiking process, largely not stimulus-specific, and hardly increased an animal's chance of choosing the correct target. Further analyses indicated that these results could marginally

improve by taking into account more neurons, but not by taking into account complementary variables such as behavioural state (see Fig 6). This suggests that in Data set 2, average population firing rates were not the central mechanism driving perceptual decision-making.

The disparity of outcomes between the two data sets examined here could be due to several factors. In many ways, Data set 1 offers an ideal case for average population responses to play a functional role: Stimulation is highly controlled, and takes place directly within the recorded brain area rather than being relayed across several synapses; lick responses are short and stereotyped, reducing movement-related neuronal dynamics; and animals are explicitly trained to detect differences in the average amount of neuronal activity within S1. In other words, even if sensory stimuli were typically not encoded in average S1 population firing rates, animals in Data set 1 may have essentially learned to 'count S1 spikes for reward'.

By comparison, Data set 2 features less controlled visual stimulation since animals can look at either of the two stimuli freely; modulation of the stimulus signal by several synaptic relays; additional neuronal dynamics driven by increased and more complex spontaneous movement in the form of wheel turning; and of course the fact that animals are free to process the difference in grating contrast in ways other than average population firing. Thus, while the two data sets examined here are clearly not sufficient to draw general conclusions, our results may point towards a scenario where cross-trial averages are more computationally relevant in settings featuring strong stimulus control and over-trained behaviours, than under more naturalistic conditions.

An alternative possibility is that we underestimated the computational relevance of cross-trial averages in Data set 2 due to idiosyncrasies of the paradigm and of our analyses. First, task-relevant stimulus information had to be computed by comparing visual inputs across brain hemispheres, but we only had access to neuronal activity from one hemisphere. Thus, recording from both hemispheres might have yielded more informative population templates. However, this data set is arguably one of the most complete sets of neuronal recordings to date regarding the number of recorded brain areas, and it therefore seems unlikely that not a single recorded brain area consistently represents integrated stimulus information from both hemifields.

A third option is that the brain areas in Data set 2 could be processing stimuli differently than in Data set 1. A popular notion is that specific features of neuronal processing may be encoded by 'super-coder' neurons that are dedicated most reliably to the feature in question. While such a segregation of neuronal sub-populations has previously been successful [48], our control analyses identified small populations of 'super-coder' neurons in only 6 out of 14 examined brain regions. Interestingly, five of those areas were sub-cortical. This might either indicate that task performance was driven crucially by sub-cortical computations, or that these sub-cortical areas are simply more likely to develop 'stereotyped' neuronal responses that repeat more accurately across trials. Moreover, taking only super-coder neurons into account actually impaired Behavioural Relevance of the resulting cross-trial averages (S10 Fig). Consistently with this, increasing the size of the overall neuronal population somewhat improved the Behavioural Relevance of cross-trial averages in both example data sets (Fig 6). Together, these findings suggest that at least in this context, cross-trial averages featuring a more eclectic mix of neuronal responses were typically more predictive of behaviour than a smaller pool of highly stable responses. This notion tallies with previous work demonstrating that large, diverse neuronal populations often convey information more efficiently than smaller, highly selective ones [24, 57, 58].

Finally, the stimulus-related response templates explored here may generally underestimate the computational power of average responses by ignoring the many stimulus and behavioural factors at play at any moment in time [5, 16, 17, 43, 45–47, 59], only some of which will be

known or accessible to the experimenter. This can make neuronal responses appear highly unpredictable, while they are actually shaped systematically and reliably by a set of unmeasured, or 'latent', variables.

We investigated this idea in two different ways. First, we grouped trials by pupil size, which is known to reflect spontaneous fluctuations in attentional state that strongly shape neuronal population activity [45, 50, 51]. Second, to account for modulating variables in a more agnostic way, we searched for distinct trial clusters that featured similar neuronal population responses. Such clusters might reflect different processing states (e.g. satiation or learning) that remain unmeasured. If either of these variables formed part of a 'multi-factorial average response curve' of the recorded neurons, then only considering trials recorded within the same attentional state or trial cluster should increase Specificity and Behavioural Relevance. This was the case in Data set 1 but not in Data set 2, suggesting that even when more latent variables are accounted for, cross-trial averages may still not be the most accurate way to reflect information processed by the brain. This does of course not preclude the possibility that test performance might improve when trials are clustered according to other ways of classifying different neurobehavioural states. For instance, one promising approach might be to define states according to overall response signatures shared by the entire neuronal population (e.g. pooled population firing rate, response burstiness or synchronization). Finally, neuronal responses may not cluster into distinct response states at all, but shift gradually over time, e.g. through learning and plasticity, stimulus adaptation or representational drift [60]. Such gradual shifts could be accounted for e.g. by including trial history as a (linear or non-linear) regressor when computing average responses.

Moreover, several recent papers have argued that factors such as stimulus properties, behavioural choices, and retrieved memories are encoded along largely orthogonal dimensions in neuronal response space [8, 61, 62]. If this is true, then extracting cross-trial averages via a dimensionality-reduction technique like PCA should significantly improve their computational relevance even in the presence of other modulating factors. This scenario held to some extent for Data set 2, where response vectors extracted by PCA became more specific though not behaviourally relevant, but not for Data set 1. It is possible that these outcomes depend on the choice of dimensionality-reduction technique. We chose PCA due to its simplicity and ubiquitous use, but other approaches like non-Negative Matrix Factorization [63] might yield different results [64]. If they were to prove more successful, this would argue in favour of analyses characterizing average neuronal response preferences simultaneously for multiple, potentially non-linearly interacting factors [65]. In this case, we would suggest that the neuroscience community abandons single-feature response averages in favour of average multi-feature response 'landscapes'. This would involve finding routine metrics to track ubiquitous latent variables like behavioural state [45, 47, 66–68] throughout a wide range of experiments.

Our results suggest that the utility of trial-averaged responses can vary dramatically across different contexts. The relevance of trial-averaging is likely to shift depending on behavioural context, stimuli, species—as well as the aspect of neuronal activity that is averaged, such as neuronal firing rates, firing phase, coherence etc. Such divergent outcomes likely hinge on the structure of cross-trial neuronal variance in a given data set. Most crucially, computationally relevant cross-trial averages can best be extracted if cross-trial variability and stimulus-driven activity operate independently of each other. However, at least in some contexts and brain areas, stimulus responses and spontaneous fluctuations of activity seem to be co-aligned [11, 43, 69–71], rendering cross-trial averages an inherently sub-optimal representation of neuronal activity in these cases. As a general recommendation, we therefore encourage researchers to study and report the variability structure of a data set (as found in its covariance matrix), as

a complementary source of information that allows to gauge the relevance of cross-trial averages.

The notion that the utility of cross-trial averages as a descriptor of neuronal activity can vary widely depending on the structure of neuronal dynamics (i.e. cross-trial variance) in a given data set was further explored in our simulations. Specifically, we modelled how Specificity and Behavioural Relevance might behave across data that contained different amounts of neuronal and behavioural variability, as well as different distributions of underlying neuronal response preferences. One should note that this model by no means aims to reflect the complexity of real neuronal activity. Instead, in the spirit of simplicity that fundamentally informs this paper, we constructed a bare-bones model that could give us an intuition of how different types of neuronal dynamics might impact our two test metrics. This allowed us to start from the test metrics observed in our example data sets, and 'backwards-induce' which sources of cross-trial variability could likely produce the observed test metrics. Somewhat surprisingly, the simulations predict that not response 'noise' but response homogeneity is the bane of computationally relevant cross-trial averages. In other words, rather than neuronal or even behavioural reliability, the diversity of neuronal response preferences is most crucial in rendering cross-trial averages relevant to neuronal computation in a given context. In line with this prediction, response preferences in Data set 1 were indeed clustered bimodally towards either one of the two stimulus conditions, while in Data set 2, responses resembled each other more uniformly.

Most importantly, we encourage researchers to compute simple 'rule-of-thumb' metrics such as the Specificity Index and Behavioural Relevance Index in order to estimate what computational role cross-trial averages play in the experimental paradigms, neuronal computations and neuronal response metrics that they study. Over time, we hope that this practice will generate a 'map' of contexts in which cross-trial averages are computationally meaningful —and motivate the field to restrict the computation of cross-trial averages to cases when they are in fact relevant to the brain.

If classical trial-averaged population responses appear largely irrelevant to ongoing neuronal computations at least in some contexts, how then could stimulus and target choice information be encoded in such cases? First, information may be encoded mostly in joint neuronal dynamics that are not captured by static (single- or multi-feature) response preferences. Analyses that take into account such dynamics, e.g. by tracking and/or tolerating ongoing rotations and translations in neuronal space [58, 68, 72, 73] or by explicitly including shared variability in their readout [74, 75], often provide vastly more informative and stable neuronal representations [58, 72]. Consistent with this, the decoder analyses (Fig 4) extracted information more successfully—most likely because decoders rely on co-variability and co-dependencies between input data and the class labels, which are smoothed over by trial-averaging.

Second, while here we have tested cross-trial averages of population firing rates as an example of basic analysis practices in neuroscience, other aspects of neuronal activity might be more informative—and as a result also potentially lead to more informative cross-trial averages. For instance, transiently emerging functional assemblies [76–78], phase relationships between neuronal sub-populations [79–81] or the relative timing of action potentials [54, 82, 83] may provide an avenue of information transmission that is entirely complementary to population firing rates.

Finally, by tracking more stimulus and behavioural variables at the same time, we can further explore how they dynamically influence overlapping and separate aspects of neuronal activity [9, 66, 67, 84–87]. Over time, we hope that this will shape our understanding of neuronal activity as an ongoing interaction rather than a static snapshot. No matter which of these approaches turns out to be most successful, it is important to recognize that time-averaged

population responses may, at least in some contexts, not be a fitting way to describe how the brain represents information. By providing a simple way to identify such contexts, the test presented here can help researchers ensure that their chosen analysis approach does the neuronal activity under investigation justice in the best possible way.

## Methods

The Methods section is divided into three subsections: Specificity and Behavioural Relevance Index, Surrogate Models and Simulations. In S1 Supplementary Methods, we expand on further subsections: Decoders, PCA analysis, Data clustering.

We have released all the scripts and data files to reproduce these analyses, they can be found at the following URL: https://github.com/atlaie/BrainAveraging. They are written in Python 3 and rely on several specialized libraries.

### Specificity Index

With the intent of characterizing whether the neural response is more similar to the appropriate template (i.e., the one corresponding to the stimulus that was actually presented in that trial) or the other one, we introduced a simple quantity we termed specificity index. It is defined as:

$$\rho_i = cor(\lambda_{correct}, r_i) - cor(\lambda_{wrong}, r_i) \tag{1}$$

where $cor$ is the Pearson correlation coefficient, $\lambda$ denotes a given neural template and $r_i$ is the population vector of the $i^{th}$ trial. Thus, the specificity index captures the differential similarity of a given neural response to each of the templates. It is key to note that, given that the Pearson correlation is bounded between −1 and 1, the specificity index can attain values between −2 and 2 and, as we were just interested in its sign and global tendencies, we did not introduce any normalization factor.

### Behavioural Relevance Index

As a way to quantify the overlap between the hit and miss distributions we used an adapted version of Vargha-Delaney's $A$ effect size [36] (also known as measure of stochastic superiority). This is an effect size derived from the Mann-Whitney U-test –a non-parametric statistical test that is particularly useful when distributions are not Gaussian [88]. Furthermore, $A$ is especially interpretable. As it is related to the $U$ statistic, it can be thought of [89] as the probability of a randomly selected point from one distribution ($X$) being higher than another randomly selected point from the other distribution ($Y$). Before computing the $U$-statistic, we have to:

1. Pool all data points into one group and sort them from low to high values.

2. Assign ranks to each sorted data point. If there is a tie (i.e., two repeated values), their rank is taken to be the average of the ranks for the entire pooled group.

3. Compute $R_X$ and $R_Y$ as the sum of the ranks of each of the groups.

   Finally, the $U$–statistic will be given by:

$$U = \min(U_X, U_Y), \tag{2}$$

with $U_X = n_X \left(n_Y + \frac{n_X+1}{2}\right) - R_X$, where $n_X$ and $n_Y$ are the number of elements in each distribution (in our case, the number of hit and miss trials, respectively), and equivalently if we flip the

*X* and *Y* labels. Having computed the U-statistic, our measure of Behavioural Relevance ($\Omega$) is given by

$$\Omega = \max(A, 1 - A), \quad \text{with } A = \frac{\frac{U_X}{n_X} - \frac{n_X + 1}{2}}{n_Y} \tag{3}$$

Thus, $\Omega$ is bounded between 0.5 and 1. If there is no overlap, $\Omega = 1$. In this extreme case, one distribution would have complete stochastic dominance over the other. If *X* and *Y* are totally overlapping, $\Omega = 0.5$ and, thus, the more its value deviates from 0.5, the less overlapping the distributions are. Note that we dropped the subscripts for $\Omega$; this is due to its definition being symmetrical for *X* and *Y*, because of the *max*($\cdot$) operation. This can be easily shown, as:

$$A_{X,Y} = 1 - A_{Y,X} \quad \Rightarrow \tag{4}$$

$$
\begin{aligned}
\Rightarrow Omega_{X,Y} &= \max(A_{X,Y}, 1 - A_{X,Y}) \\
&= \max(A_{X,Y}, A_{Y,X}) \\
&= \max(1 - A_{Y,X}, A_{Y,X}) \\
&= \Omega_{Y,X}
\end{aligned}
\tag{5}
$$

Thus, the definition of the Behavioural Relevance index is independent of the order in which we take the statistical test.

## Surrogate models

**Calcium imaging data.** For this data set, we pooled together those trials within the same stimulus set: on the one hand, when stimulation was given; on the other hand, those trials without any stimulus. For each of those trial groups, and for each neuron, we built a Gaussian distribution with mean and variance given by the trial-average and trial-variance. We then sampled 200 random values for each stimulus set and correlated each of them with the template.

**Spike data.** We were interested in comparing the experimental neuronal population response with a downsampled version of the trial-averaged template. To do that, we built our surrogate models by constructing *N*(= 100) random vector with the following constraints:

1. Its size is equal to the number of neurons comprising the neural population for that area and that session.

2. The probability that at n spikes are allocated at a particular location m (i.e., that neuron m has spiked n times) is given by $P_{m,n} = \left(\frac{\lambda_m}{\sum_m \lambda_m}\right)^n$, where $\lambda_m$ is the $m^{th}$ element of the template vector.

3. The total number of spikes is constant and equal to the total recorded number of spikes for that area and that session.

By imposing these constraints, we are testing the alternative hypothesis that neurons are independent from each other (uncorrelated) and it is therefore equivalent to keeping the single-trial population statistical response, while scrambling across trials. This is also the same as drawing single-neuron responses from the underlying template distribution following a Poisson process. Thus, the bootstrapped responses contain the same number of spikes as the original trial, but the neurons that produced these spikes are randomly chosen according to their probability of occurrence in the average template.

**Yule-Kendall index.** In order to assess how symmetrical the jackknifed distributions shown in S6 Fig are, we relied on the Yule-Kendall index, which is computed as:

$$\gamma = \frac{Q(3/4) + Q(1/4) - 2Q(1/2)}{Q(3/4) - Q(1/4)}, \tag{6}$$

where $Q$ is the quantile function. We chose this measure because it works for non-normal distributions and because it is non-dimensional (thus allowing direct comparison between data sets).

**Simulations.** In order to systematically explore the two assumptions that we tested in both experimental data sets, we implemented a simple model. We begin by generating two stimuli, denoted as $S_1$ and $S_2$. We assumed, for simplicity, that they can only attain two possible values (0 and 1). For each of the stimuli, a series of trials are generated. Each trial is an independent and identically distributed draw from the Bernoulli distribution. Thus, for each stimulus $S$ with parameter $p$, the probability mass function is:

$$P(S = k) = p^k(1 - p)^{1-k}, \quad k \in \{0, 1\} \tag{7}$$

Where: $P(S = k)$ is the probability that $S$ takes the value $k$, $p$ is the probability of drawing that value and $(1 - p)$ is the probability of drawing the other one.

In this simulation, we chose $p_1 = p_2 = 0.5$, resulting in two possible equally likely stimuli.

Then, we simulated the activity of a population of neurons in response to different stimuli. We assumed that each neuron $j$ has a certain baseline firing rate ($r_i^0$) and a certain stimulus gain ($g_j^s$), both of them sampled from the normal distribution. Both distributions, are given by $\mathcal{N}(5, 1)$. Moreover, each neuron has an intrinsic selectivity value ($\xi_j$), which determines how the neuron's firing rate changes depending on the difference between the two stimuli. The selectivity values were generated from a Beta distribution, controlled by the parameter $\beta$.

Specifically, let $\xi$ denote the vector of selectivities all neurons, with $\xi_i$ indicating the selectivity of neuron $i$. Then, each $\xi_i$ is generated as follows:

$$\xi_i \sim Beta(\beta, \beta) \tag{8}$$

The selectivity values are then shifted and scaled to lie within $[-1, 1]$.

For each trial, the firing rate of each neuron depends on its selectivity and the difference between the stimuli. The difference between the stimuli ($\Delta S$), is computed for each trial, resulting in a sequence of stimulus differences.

The firing rate for each neuron $j$ on trial $i$ is then calculated as follows:

$$r_{i,j} = ReLU(r_j^0 + g_j^s \cdot \xi_j \cdot \Delta S_i) + \left|\mathcal{N}(0, 1)\right| \tag{9}$$

where $r_j^0$, $g_j^s$ and $\xi_j$ are the baseline firing rate and stimulus gain the selectivity for neuron $j$; $\Delta S_i$ is the stimulus difference in trial $i$. Then, we pass the computed firing rate through a ReLU in order to add a simple non-linearity and to ensure positivity.

Finally, the firing rates are used to generate spikes following a Poisson distribution. For each neuron $j$ on a trial $i$, the number of spikes, denoted as $s_{i,j}$, is generated as:

$$s_{i,j} \sim Poisson(r_{i,j}) \tag{10}$$

After we had the population response to the presented stimulus, we proceeded as before: we computed a "left" and "right" template (i.e., $\Delta S = 1$ or $\Delta S = -1$); then, for a single-trial, we computed the similarity of the population vector in that trial with each of the templates ("correct" and "wrong"). The difference in between these is the Specificity Index.

In order to model the decision-making process, we assumed that the relevant quantity to make a decision was the Specificity Index, as it follows from the template-matching procedure. We assumed there was a noisy process on top of the template-matching. Then, for a given trial, the noisy Specificity Index ($\psi'$) is simply:

$$\psi' = (\rho_{correct} - \rho_{wrong}) + N(0, q^2) \tag{11}$$

Then, and the corresponding outcome is given by:

$$outcome = \begin{cases} 0 & \text{if } p < 0.5 \\ 1 & \text{if } p \geq 0.5 \end{cases} \quad p = \frac{1}{1 + e^{-\psi'}}, \tag{12}$$

Thus, if the probability value is greater than 0.5, that trial is considered a hit (1); otherwise, it is a miss (0).

Crucially, we added an stochastic component into the decision-making, which means that even for trials with the same Specificity Index, different decisions can be made.

## Supporting information

**S1 Supplementary Methods. These Supplementary Methods are divided into three subsections: I) Decoders; II) PCA analysis; and III) Data Clustering.** We detail all the relevant parameters and tools we made use of.
(PDF)

**S1 Fig. Example of neuronal responses for both Data sets.** A) For Data set 1, we show a session, for all neurons over time, aligning the responses to stimulus onset (pink). Traces of individual cells are depicted in light gray, their trial-average in black. In cyan, we show the analysis window (500*ms*), within which we take the time-average (colored dots; blue for S1, orange for S2) that we use to construct the population vectors we used for the analyses. B) Spike traces for different representative areas in Data set 2 (a stimulus informative one [blue, primary visual cortex, VISpm], a choice-informative one [red, ventral anterior-lateral complex of the thalamus, VAL] and a both-informative one [purple, anterior cingulate area, ACA]). Each one comes from a different session and has a different number of neurons. As before, in pink we show the stimulus presentation and in cyan the analysis window (200*ms*) that we used to construct the population vectors in the analyses.
(EPS)

**S2 Fig. Correlations between response templates for different stimulus constellations.** A) For Data set 1, we pool together those trials with a low number of stimulated neurons ($n \in [5, 20]$) and compare the trial-averaged response with those trials with a higher number of stimulated neurons ($n \in [30, 50]$). Their correlation generally exceed 0.6, suggesting that one template should be sufficient to represent different contrast constellations. B) For Data set 2, we compute the similarity between templates (for each contrast level: 0.25, 0.5, 1, referred to as low, mid and high, respectively) for each screen (left, right). Their correlation is generally above 0.8, so grouping them together should provide a coherent representation.
(EPS)

**S3 Fig. Single-trial responses are not stimulus-specific for Data set 2.** A) Distribution of the correlations between single-trial responses and the matching trial-averaged response templates. The top diagram shows the criterion to "match" single-trial responses with the trial-average. Box: $25^{th}$ and $75^{th}$ percentile. Center line: median. Whiskers: $10^{th}$ and $90^{th}$ percentile.

Dotted lines: median of bootstrapped data. B) Same as A) but for the non-matching combinations. C) Distribution of the correlations between single-trial responses and the matching trial-averaged response templates, split by behavioural outcome. It can be seen that these distributions are practically identical for all regions.
(EPS)

**S4 Fig. Representation of the bootstrapping procedures.** A) In the first data set, we created the templates based on the time-average for each neuron, for each trial. We then constructed a Gaussian distribution centered around the trial-average and with a spread equal to the trial-variance. We repeated this for each neuron and then sample 200 times for each stimulus set. B) For data set 2, we fixed the number of spikes on each trial, but randomly assigned to a neuron with a probability according to how often it spiked in the average template. We repeated this procedure 100 times.
(EPS)

**S5 Fig. Reaction time distribution.** General distribution for reaction times, over all sessions, split by hits and misses. Consistent with the literature, misses significantly (Mann-Whitney's U-test, $p = 1.88 * 10^{-172}$, $A = 0.68$) imply longer reaction times. The vertical line indicates the width of the time window in which we have performed all of our analyses ($200ms$). We have selected our analysis window of because of its likely relevance to stimulus processing and behavioural decision making.
(EPS)

**S6 Fig. Population variability is not predictive of stimulus-specificity.** A) Scatter of the Specificity Index for different population variances. There is virtually no correlation for both regions. B) Fano factor distribution over all recorded regions. As for Data set 1, variability is uncorrelated with the Specificity Index and it is, thus, not capturing the same effect.
(EPS)

**S7 Fig. Specificity Index and Behavioural Relevance for all areas.** Same as Fig 5B and 5E, but without pre-selecting informative areas. Results are preserved in general: across areas, single-trial responses are barely stimulus-specific (Specificity Indices are tightly clustered around 0, with a median value of 0.019) and not very behaviourally relevant (median value of $\Omega$ = 0.57).
(EPS)

**S8 Fig. Easy trials are more stimulus-specific and behaviourally relevant than difficult trials for Data set 2.** A) Task structure and behavioural proficiency; easy trials are taken to be the extreme cases: low contrast shown on one screen and a high one in the other. B) Distribution of the Specificity Index for easy and difficult trials, over the selected areas. In color, significant comparisons. Box: $25^{th}$ and $75^{th}$ percentile. Center line: median. Whiskers: $10^{th}$ and $90^{th}$ percentile. Dotted lines: median of bootstrapped data. C) Behavioural relevance ($\Omega$) for easy and difficult trials, over all selected areas. D) Same as B) but for all recorded areas. E) Same as C), but for all recorded areas. Average responses are more behaviourally relevant for easy than for difficult trials (right).
(EPS)

**S9 Fig. Jackknife analyses.** We remove one neuron at a time and compute the impact in the Specificity Index, calculated as $SpecIdx_i^{JK} = n \cdot SpecIdx - (n - 1) \cdot SpecIdx^{-i}$, for neuron $i$. We show the symmetry ($\gamma$) of each distribution around 0 using Yule's coefficient (Methods). This measure is bounded between −1 and 1, with each of them meaning completely skewed towards negative or positive values, respectively. A) For both areas in Data set 1, change in single-trial

Specificity Index when one individual neuron was removed. Data points: Trials. Colors: see inset legend. B) Same for Data set 2.
(EPS)

**S10 Fig. Extremely selective neurons.** We select the top (bottom) 10% most informative neurons (as given by their Jackknifed Specificity Index). We then plot how the Specificity Index and Behavioural Relevance look like (compared to selecting the entire population, taken as baseline) when only including these most (least) informative neurons. A) Perhaps unsurprisingly, the Specificity Index increases markedly (decreases slightly) when selecting the most (least) informative neurons only. B) There is no consistent improvement of the Behavioural Relevance when we just select one group or the other. Data points: Trials. Colours: more saturated indicates "most informative neurons"; less saturated, "least informative neurons". B) Same for Behavioural Relevance.
(EPS)

**S11 Fig. Unsupervised trial-clustering for both data sets.** Silhouette Index (SI) for both Data sets. This quantity measures cluster compactness (see Methods), with 0 indicating complete overlap between spread clusters and 1 meaning perfectly separated ones. Different colors represent a different number of clusters, from $k = 2$ to $k = 5$. A) SI for Data set 1. In this case, clusters are not compact and they are better distinguishable (for both brain areas) when $k = 2$. B) SI for Data set 2. Clusters are more compact than in the previous case, with $k = 2$ also being the best option. Thus, for analyses in the main text, we decided to group trials into two clusters.
(EPS)

**S12 Fig. PCA is not helpful in making single-trial responses more stimulus-specific or behaviorally relevant for Data set 2.** A) Distribution of single-trial normalized distances in PCA space (Methods) between response vectors and the trial-averaged response template for the correct (top) and wrong (bottom) stimulus constellation. B) Same as B, but the Specificity index of single-trial responses across brain areas, defined as the difference between the normalized distance to the wrong and correct template. Solid gray line highlights the Specificity Index of 0.0, which translates to exactly equal correlation to correct and wrong template. Dotted lines represent the specificity index of the medians of the bootstrapped values for each recorded area. C) Same as A (top), split by hits and misses. These distributions almost completely overlap, for all brain areas. D) Same as B, split by hits and misses. As in the previous panel, these distributions are practically coincident. E) Behavioural Relevance Index for all brain areas. They range from 0.50 to 0.62, with $\Omega = 0.5$ meaning perfect overlap.
(EPS)

**S13 Fig. PCA is helpful in making single-trial responses more stimulus-specific but heavily reduces behaviourally relevant for Data set 1.** A) Same as Fig 2 B) Same as Fig 3C. In this case, the Specificity Index increased modestly (from 0.13 to 0.21) at the expense of severely hindering behavioural relevance (from 0.83 to 0.52).
(EPS)

**S14 Fig. Selectivity distributions for both Data sets.** Data set 1 (shown in A) shows generally higher, and more uniform distributions of, neural selectivity than Data set 2 (shown in B). In Data set 2, the majority of neurons are barely selective, and the distribution is similar to a normal distribution ($\beta \gg 1$). Only some neurons are extremely selective, which is in line with the previous discussion of S10(A) Fig. It is also worth noting that in Data set 1, the distribution for S1 is visibly more skewed than S2; this suggests that already a few synapses away from the

stimulation site, selectivity decreases markedly.
(EPS)

## Acknowledgments

We thank Jonathan Pillow, Viola Priesemann, Nick Steinmetz, Cyrille Rossant, Miles Wells, Joshua Gold and Mike X Cohen for valuable input on earlier versions of the manuscript.

## Author Contributions

**Conceptualization:** Alejandro Tlaie, Katharine Shapcott, Adam Packer, Paul Tiesinga, Marieke L. Schölvinck, Martha N. Havenith.

**Data curation:** Thijs L. van der Plas, James Rowland, Robert Lees, Joshua Keeling, Adam Packer.

**Formal analysis:** Alejandro Tlaie.

**Funding acquisition:** Marieke L. Schölvinck, Martha N. Havenith.

**Investigation:** Alejandro Tlaie, Katharine Shapcott, Marieke L. Schölvinck, Martha N. Havenith.

**Methodology:** Alejandro Tlaie, Thijs L. van der Plas, Adam Packer, Paul Tiesinga, Marieke L. Schölvinck.

**Project administration:** Alejandro Tlaie, Marieke L. Schölvinck, Martha N. Havenith.

**Resources:** Adam Packer, Martha N. Havenith.

**Software:** Alejandro Tlaie, Thijs L. van der Plas.

**Supervision:** Marieke L. Schölvinck, Martha N. Havenith.

**Visualization:** Alejandro Tlaie.

**Writing – original draft:** Alejandro Tlaie, Marieke L. Schölvinck, Martha N. Havenith.

**Writing – review & editing:** Alejandro Tlaie, Katharine Shapcott, Thijs L. van der Plas, Adam Packer, Paul Tiesinga, Marieke L. Schölvinck, Martha N. Havenith.

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
