## [Decision Letter · Decision Letter 0]

12 Dec 2023

Dear Mr Tlaie Boria,

Thank you very much for submitting your manuscript "What does the mean mean? A simple test for neuroscience" for consideration at PLOS Computational Biology. As with all papers reviewed by the journal, your manuscript was reviewed by members of the editorial board and by several independent reviewers. The reviewers appreciated the attention to an important topic. Based on the reviews, we are likely to accept this manuscript for publication, providing that you modify the manuscript according to the review recommendations.

While normally we would ask for a revised manuscript within 30 days, we understand that this overlaps with a common long holiday period so if you anticipate it taking longer than 30 days, please let us know the expected resubmission date by replying to this email.

Sincerely,

Bradley Voytek

Guest Editor

PLOS Computational Biology

Lyle Graham

Section Editor

PLOS Computational Biology

Reviewer's Responses to Questions

**Comments to the Authors:**

Reviewer #1: Tlaie and colleagues investigate novel criteria for defining brain areas as relevant for a task: reliability of the responses and a relationship between single-trial consistency and behavioral responses. The authors use two datasets for their case study, and analyze these datasets rigorously and in depth. Dataset 1 passes the criteria but dataset 2 does not pass these criteria because, the authors claim, the information is dynamically encoded. However, I think this is an example of the short-comings of the criteria, particularly criterion 1. I discuss the caveats of the approach below, and note some potential discussion points.

The single-trial responses are in most cases positively correlated to the trial-averaged responses to some extent. The close-to-zero correlations observed on many trials are consistent with low firing rates with poisson variability. Alternatively, one extreme example I can think of where the correlation would be low is if there are two populations of neurons that are driven by a given stimulus on average, but on any given stimulus one set of these stimulus-selective neurons is on and the other is suppressed somewhat. Both of these cases do not reduce the ability of a downstream area to decode information: a downstream brain area can decode the stimulus by pooling the stimulus-driven neurons’ inputs.

This for example is the function of the “super-neurons” in (Stringer et al 2021), to pool over many neurons, not to choose the “super-coder” neurons. Indeed, the authors found that the behavioral relevance criterion increases as a function of the number of neurons considered. Also, in (Perez-Ortega et al 2021) neuron groups are found using unsupervised clustering (not directly using stimulus-driven properties), and there is no mention of finding “super-coder” neurons; instead they discuss the stability of these groups of neurons across days.

A case where the decoder *would* fail is if the trial-to-trial variability was information-limiting (Moreno-Bote et al 2014). For example, if in response to the left grating, all the neurons responded on some trials like the grating was on the right, then the stimulus could not be decoded from the population on those trials. A neural population can be unreliable but not have these detrimental correlations. Indeed using dataset 2 the authors claim that motor and visual areas are “barely reliable and behaviorally relevant”, but inactivation of these areas leads to the mouse being unable to perform the task (Zatka-Haas et al 2021). Thus, the claim that these measures, particularly reliability, relate to task performance is unproven.

I still think these criteria are very useful, just with some of these caveats mentioned. For example, it might be useful to discuss how increased numbers of neurons improve the behavioral relevance, suggesting that sampling more neurons per brain area can influence criterion 2. Also, it might be useful to discuss that certain types of unreliability are more detrimental than others to decoding analyses.

Reviewer #2: Review: What does the mean mean? A simple test for neuroscience

In their manuscript, Tlaie and co-authors embark on an exciting crusade to uncover the mean and meaninglessness of the trial-averaged view of population activity. They propose two central measures to capture how informative the averages are to characterize population activity and to predict behavioral outcomes. They start from the first principles:

1. If means are informative, they shall be well correlated with the population activity at each trial, and more so, they should be more correlated with the average activity of the same trial condition than with the different one.

2. If the upstream population (or researcher) is using means as a perfect template for the behavioral choice, closeness to the mean should be predictive of the quality of the behavioral response.

The authors derived the measures to capture how well these two principles are realized in the recorded data and evaluated them on two datasets. Interestingly, they found that in one dataset (with more constrained experimental conditions), the means were a rather good description of the population activity. In contrast, in the less constrained conditions, they were uninformative.

I generally liked the paper: the question is relevant and timely, the methods are solid and imaginative, the images are great, and the writing is quite clear.

I have some moderately minor and some really minor comments.

To me, it feels that results are sometimes stated stronger than what is visible from the images/analysis. Specifically, in the discussion of 5B. and C., the authors state in 208-209, “in Data set 2, the relation between single-trial responses and trial-averaged 209 templates is neither reliable nor specific.” However, in the figure and discussion, they need to use arguments that the indices/differences are significant but not large enough. While I support the usage of effect sizes, I feel they do not warrant, in this case, such a strong statement.

Further on, figure 7A: Here, I am not sure if the expected specificities of different data pre-processing are of the same ranges. In any case, it would be interesting to read why the reliability of the z-scored firing rate is larger than for Firing rates in Data2 and smaller in Data 1, and if the reliability of some data pre-processing is so large, is it still reasonable to say that the means are not informative for data2?

A similar problem seems to appear in the discussion of 6C: 280: “… grouping trials by average pupil size did not improve specificity”. However, panel 6C on the Specificity index clearly shows an increase, even if possibly not sufficient. Even more so for 6B and specificity (we expect it to grow because of picking the more correlated neurons, but the statement seems to contradict the picture)

The part on the surrogate data model is hard to read, the main text didn’t call the distribution (Beta distribution) and didn’t reference the methods. It refers to the function as Gaussian (299), but it just looks bell-shaped, it has a constrained domain so cannot be Gaussian. For Fig 8 it is unclear where the squares indicating data comes from. In the suppl Fig. it does not look like the experimental data is symmetric, so would not be a uni-parametric Beta distribution. Did you fit Beta to responses? How was SNR evaluated for the data?

Some writing comments:

The paper is overall very well written. My suggestions are mainly to improve the ”reader experience”.

I was missing the references to the particular parts of the vast method section. I feel they are also necessary because, at least for me, it was challenging to match the section titles with the information I sought. I would suggest adding numbering/lettering to the subsections in methods and referring to them directly.

For me, some parts feel a bit bloated, making it harder to get to the (simple and straightforward) point. Notably, the discussion repeats the results in a rather long form. I would suggest shortening it, especially the parts about checking alternative hypotheses.

Also, in the discussion (366-372), you mention “benchmarking.” This term has a strong meaning in some communities, overlapping with PLOS CB readership (checking the performance on a standardized test and comparing it to the previously presented solutions). I think this meaning does not fit your case. You illustrate the usage of the metrics, but you do not have ground truth or competitor-metrics.

Minor:

In Fig 2C middle: it seems like the bootstrap is still form the correct trial group. If not than I missed how it is done and why you have still the same value as on the left.

For Fig 4B-C, it feels like the thresholds for B were taken such that the areas would end up being exactly same as in C (at least for the Stimulus Information even the smallest threshold change will add or remove some areas). I think the match is remarkable independent on exact thresholding but maybe the total coincidence invites over-confidence in the results. For the better visibility, I would have colored the names of the most informative areas in C in the colors corresponding to their counterparts in B.

Line 187: I think you might have meant S3A

298 and later in the text something happened with “<” and “>”

314 “Data Set 1 should” be distributed

In methods: 617 you mean the “Pearson correlation coefficient”.

In Behavioral relevance part would be good to tell what are n_X, n_Y

Eq 17: k in {0,1} (you need brackets for the correct notation)

Fig S12. plotting (as, e.g., non-filled circles) the Omega for raw firing rates would make your statement visible

Fig S13 lacks the caption.

**Have the authors made all data and (if applicable) computational code underlying the findings in their manuscript fully available?**

Reviewer #1: Yes

Reviewer #2: Yes

PLOS authors have the option to publish the peer review history of their article (what does this mean?). If published, this will include your full peer review and any attached files.

Reviewer #1: No

Reviewer #2: No

Figure Files:

Data Requirements:

Reproducibility:

References:

---

## [Decision Letter · Decision Letter 1]

12 Mar 2024

Dear Mr Tlaie Boria,

We are pleased to inform you that your manuscript 'What does the mean mean? A simple test for neuroscience' has been provisionally accepted for publication in PLOS Computational Biology.

Best regards,

Bradley Voytek

Guest Editor

PLOS Computational Biology

Lyle Graham

Section Editor

PLOS Computational Biology

Reviewer's Responses to Questions

**Comments to the Authors:**

Reviewer #1: Thank you to the authors for thoroughly addressing my comments, and clarifying some points which I misunderstood. I think the manuscript is much clearer and is going to be useful for neuroscientists to read.

Reviewer #2: In the revised version of their manuscript “What does the mean mean? A simple test for neuroscience”, Tlaie and co-authors addressed all my concerns. The changes made to the paper are sufficiently reflecting the responses and the manuscript now is ready for publicaition.

**Have the authors made all data and (if applicable) computational code underlying the findings in their manuscript fully available?**

Reviewer #1: Yes

Reviewer #2: Yes

PLOS authors have the option to publish the peer review history of their article (what does this mean?). If published, this will include your full peer review and any attached files.

Reviewer #1: No

Reviewer #2: No

---

## [Editor Report · Acceptance letter]

2 Apr 2024

PCOMPBIOL-D-23-01578R1 

What does the mean mean? A simple test for neuroscience

Dear Dr Tlaie Boria,

I am pleased to inform you that your manuscript has been formally accepted for publication in PLOS Computational Biology. Your manuscript is now with our production department and you will be notified of the publication date in due course.

With kind regards,

Olena Szabo
